# Current State of Human Gene Therapy: Approved Products and Vectors

**DOI:** 10.3390/ph16101416

**Published:** 2023-10-05

**Authors:** Aladdin Y. Shchaslyvyi, Svitlana V. Antonenko, Maksym G. Tesliuk, Gennadiy D. Telegeev

**Affiliations:** Institute of Molecular Biology and Genetics, National Academy of Sciences of Ukraine, 150, Zabolotnogo Str., 03143 Kyiv, Ukraine; s.v.antonenko@imbg.org.ua (S.V.A.); m.g.tesliuk@imbg.org.ua (M.G.T.); g.d.telegeev@imbg.org.ua (G.D.T.)

**Keywords:** human gene therapy, gene therapy drugs, viral vectors, non-viral vectors, physical delivery methods, human gene therapy products

## Abstract

In the realm of gene therapy, a pivotal moment arrived with Paul Berg’s groundbreaking identification of the first recombinant DNA in 1972. This achievement set the stage for future breakthroughs. Conditions once considered undefeatable, like melanoma, pancreatic cancer, and a host of other ailments, are now being addressed at their root cause—the genetic level. Presently, the gene therapy landscape stands adorned with 22 approved in vivo and ex vivo products, including IMLYGIC, LUXTURNA, Zolgensma, Spinraza, Patisiran, and many more. In this comprehensive exploration, we delve into a rich assortment of 16 drugs, from siRNA, miRNA, and CRISPR/Cas9 to DNA aptamers and TRAIL/APO2L, as well as 46 carriers, from AAV, AdV, LNPs, and exosomes to naked mRNA, sonoporation, and magnetofection. The article also discusses the advantages and disadvantages of each product and vector type, as well as the current challenges faced in the practical use of gene therapy and its future potential.

## 1. Introduction

Gene therapy has become a rapidly growing field with significant advancements in recent years. This innovative therapeutic approach is revolutionizing the treatment of various diseases, including melanoma, pancreatic cancer, retinal dystrophy, spinal muscular atrophy, polyneuropathy, hereditary transthyretin-mediated amyloidosis, head and neck squamous cell carcinoma, atherosclerotic peripheral arterial disease, critical limb ischemia, nasopharyngeal cancer, SOS/VOD with multiorgan dysfunction, bacillus Calmette–Guérin (BCG)-unresponsive non-muscle invasive bladder cancer, hemophilia B, aromatic L-amino acid decarboxylase deficiency, multiple myeloma, cerebral adrenoleukodystrophy, lymphoma, ADA-SCID, large B cell lymphoma, acute lymphoblastic leukemia, beta-thalassemia, and metachromatic leukodystrophy, by addressing the root cause of the condition—the genetic level. By altering, repairing, or replacing defective genes in a patient’s body, gene therapy aims to restore the normal functioning of cells and tissues. Since the first gene therapy clinical trial in 1989, significant progress has been made in developing and refining the technology, leading to the approval of several gene therapy products for commercial use [1,2].

Delving into the annals of gene therapy’s history unveils a narrative shaped by pioneering discoveries and watershed moments. Frederick Griffith’s 1928 experiment resonates as a seminal exploration, elucidating the transference of genetic information among bacteria through the transformative process. A monumental breakthrough arrived in 1972 with Paul Berg’s revelation of the first recombinant DNA, a milestone that would reverberate across the scientific community. Berg’s paradigm-shifting achievement garnered him the 1980 Nobel Prize in Chemistry, underscoring the transformative power of recombining DNA molecules. In a pivotal moment on 19 January 1989, Dr. James A. Wyngaarden, director of the National Institutes of Health (NIH), granted approval for the inaugural clinical protocol, entailing the integration of a foreign gene into immune cells for cancer patients. On 14 September 1990, another indelible moment occurred, as W. French Anderson and his NIH colleagues conducted the first sanctioned gene therapy procedure, a life-altering intervention for a four year old afflicted with severe combined immunodeficiency (SCID). From 1990 to 2000, the landscape burgeoned with promise as approximately 300 clinical gene therapy trials embraced around 3000 individuals, marking a dynamic era of exploration. The march of progress ventured eastward as China’s State FDA granted approval for Gendicine in 2003 to address carcinoma and Oncorine in 2005 for nasopharyngeal cancer. Russia followed suit, gaining approval for Neovasculgen in 2011 to combat atherosclerotic peripheral arterial disease (PAD). The historical trajectory surged ahead, with the European FDA approving Defibrotide in 2013 for addressing SOS/VOD with multiorgan dysfunction. In 2015, the USA FDA’s imprimatur ushered in a new era with IMLYGIC, ratified to treat melanoma and pancreatic cancer. The progression was resounding, as 2023 bore witness to the culmination of these endeavors, witnessing the approval of 22 gene therapy products by entities such as the USA’s FDA, the EU’s FDA, the Chinese State’s FDA, and the Russian Ministry of Healthcare, emblematic of the multifaceted strides taken to harness the potential of gene therapy on a global scale [3,4,5,6].

Examining gene therapy from a broad perspective involves two critical components: genetic drugs (also referred to as passengers, which are technicians capable of repairing a malfunction) and carriers of the drugs (or vehicles capable of delivering the technicians to the point on the navigational map where the repair job can be performed).

Each drug and carrier have their advantages and drawbacks, and there is no one-size-fits-all solution. Therefore, the more effectively the pros and cons of the combinations of these two elements are used, the more successful the drugs are in reaching their destination and producing beneficial results.

Unfortunately, some combinations of these two elements are unable to reach their intended destination, while others get there, but the cost of repairing becomes prohibitively high, resulting in the rejection of the offer by the market. Only about 11.5% of these combinations succeed after years of experimentation and remedying failures; these are the gene therapy products that have been authorized by government authorities and embraced by the market demand. In this article, we provide an overview of the current state of gene therapy, focusing on approved products (Figure 1) [1,7,8].

## 2. Sixteen Gene Therapy Drugs

Gene therapy drugs have revolutionized the field of medicine by providing a targeted approach to treating genetic disorders. In this section, and in Table 1, we will provide an overview of various gene therapy drugs that have been developed, their advantages and disadvantages, as well as the challenges faced in their practical use.

### 2.1. Small Interfering RNA (siRNA)

siRNA is a synthetic molecule used to knock down the expression of any gene with a complementary sequence. The molecule works by targeting specific mRNA and cleaving it, preventing it from being translated into protein. Additionally, siRNA can also be used to regulate protein-coding genes and transposons, as well as functioning as an antiviral defense mechanism.

The size of siRNA ranges from 20 to 25 base pairs. One of the key advantages of siRNA is its high specificity due to its 100% complementarity to the target mRNA. This makes it an attractive drug candidate for diseases caused by specific gene mutations. Additionally, siRNA has been shown to be effective in delivering drugs to the brain, a feat that is notoriously difficult to achieve.

However, siRNA therapy also has some limitations. One major concern is off-target effects, which means that the siRNA may unintentionally target genes with similar sequences to the intended target. Another potential issue is innate immunity, which can cause an immune response and limit the effectiveness of the therapy [9,10,11,12,13,14,15].

### 2.2. MicroRNAs (miRNAs)

miRNAs are short non-coding RNA molecules that regulate gene expression at the post-transcriptional level. They play a significant role in a wide range of cellular processes, including differentiation, apoptosis, and development. In plants, miRNAs and their target mRNA are almost perfectly complementary, making them highly effective. They are involved in developmental timing, tissue growth, and left–right asymmetry in the nervous system. In animals, miRNAs comprise only approximately 1% of all genes, but they play an essential role in regulation, including mRNA degradation, translational repression, and the regulation of protein-coding genes.

MicroRNAs (miRNAs) offer a key advantage in gene therapy due to their small size and manipulability. Moreover, around 12 miRNAs have been identified for suppressing endogenous CFTR mRNA expression in the Caco-2 cell line. CFTR, responsible for the monogenic autosomal recessive cystic fibrosis (CF), impacts 1 in 3500 global live births. Maria V. Esposito et al. [76] examined 706 CF carriers, revealing undiagnosed CFTR-RD among a subset. Genetic testing scanning analysis aids in CFTR-RD identification, offering potential for tailored follow-up and therapies to enhance outcomes.

However, functional duplexes in animals can be more variable in structure than in plants, with only short complementary sequence stretches that may contain gaps and mismatches. Specific rules for functional miRNA–target pairing that capture all known functional targets have not been developed to date [11,16,76,77,78].

### 2.3. PIWI-Interacting RNAs (piRNAs)

piRNAs are small non-coding RNA molecules that interact with PIWI proteins to repress transposable elements in the genome.

piRNAs are known to have diverse functions such as transcriptional or post-transcriptional repression of transposons and multigenerational epigenetic phenomena in worms. In addition to transposon silencing, pre-pachytene piRNAs also have roles in the formation of the nuage, a perinuclear structure in germ cells.

These RNAs are larger than other small RNAs, typically ranging from 26 to 32 nucleotides in length.

While the exact mechanisms of piRNA biogenesis remain unclear, current models suggest that they are processed from long, single-stranded RNA precursors in a Dicer-independent manner. Studies on piRNAs are still ongoing to understand the full range of their functions and mechanisms of action [15,17].

### 2.4. Short Hairpin RNA (shRNA)

shRNA is an artificial RNA molecule used for gene silencing via RNA interference. This type of drug contains a hairpin turn that tightly binds to its target gene, leading to suppression of its expression.

shRNAs range in size from 19 to 29 base pairs and have the advantage of being relatively resistant to degradation and turnover, providing long-lasting gene silencing effects.

However, to use shRNA, an expression vector is required, which may cause side effects when used as a medicine.

Despite these limitations, shRNA is considered an effective tool for gene therapy due to its specific targeting ability and long-term effects [18,19,20,21,22].

### 2.5. Antisense Oligonucleotides (ASOs)

ASOs are a type of drug that have gained significant attention in recent years due to their potential in gene therapy. ASOs are single strands of DNA or RNA that are complementary to a specific sequence of mRNA. They work by binding to the targeted RNA and blocking the translation of certain proteins, thereby modulating gene expression. ASOs are relatively small in size, ranging from 18 to 30 base pairs, which enables them to easily penetrate cell membranes and target both nuclear- and cytoplasmic-located long non-coding RNAs (lncRNAs).

Despite their potential therapeutic benefits, ASOs have several limitations that need to be addressed. One major concern is off-target effects, where ASOs bind to unintended RNA sequences and cause unwanted biological effects. Additionally, ASOs may have insufficient biological activity, limiting their efficacy in gene therapy.

ASOs have shown promising results in clinical trials for treating genetic disorders, such as spinal muscular atrophy and Huntington’s disease, and several ASOs have been approved by the FDA [23,24,25,26,27,28,29].

### 2.6. Oligodeoxynucleotides (ODNs)

ODNs are synthetic DNA molecules that have shown potential as a gene therapy tool. ODNs work through two main mechanisms: the antisense strategy and the antigene strategy (also known as the decoy strategy). In the antisense strategy, ODNs bind to the targeted mRNA and block protein synthesis, while the antigene strategy involves the use of ODNs to bind to specific transcription factors and inhibit their activity, thus preventing the expression of downstream genes.

One of the benefits of using ODNs is the simplicity of their synthesis and manipulation, as well as the tissue specificity of their target transcription factors. This specificity enables precise targeting of specific genes or cell types, reducing the risk of off-target effects.

However, ODNs also have several limitations that need to be addressed. One major concern is their high rate of degradation by endocytosis or nucleases, which limits their stability and effectiveness. Additionally, their short lifetime may reduce the duration of therapeutic effects.

Despite these limitations, ODNs have shown potential in preclinical and clinical trials for treating various diseases, such as cancer and autoimmune disorders [30].

### 2.7. Clustered Regularly Interspersed Short Palindromic Repeats (CRISPR)/CRISPR-Associated Protein 9 (Cas9)

In recent years, CRISPR/Cas9 has been making waves in the field of gene therapy. This protein, formerly known as Cas5, Csn1, or Csx12, plays a critical role in the defense of certain bacteria against DNA viruses and plasmids. Its primary function is to cut DNA, which allows for the alteration of a cell’s genome.

One of the biggest advantages of CRISPR/Cas9 is its ease of design. It can be delivered via plasmid or viral vectors, which makes it accessible to many researchers. However, there are some downsides to consider. For example, off-target editing is common without an additional homologous sequence, which can be a challenge.

Another drawback is the requirement for a PAM or Protospacer Adjacent Motif, a short sequence downstream of the target DNA sequence. While this motif is necessary for CRISPR/Cas9 to work, it can also limit the range of targets available for editing [31,32,33,34,35].

### 2.8. Plasmid DNA (pDNA)

One of the most promising vectors for gene therapy is pDNA. These small, extrachromosomal DNA molecules replicate independently and are physically separated from chromosomal DNA within a cell. pDNA gene therapy has been shown to be particularly effective in the treatment of cardiovascular diseases because it allows for targeted transfer to cardiac or skeletal muscle.

One of the major advantages of using pDNA as a vector is its versatility in size. This allows for a wide range of genes to be delivered using this method. Additionally, plasmids can be engineered to include a variety of promoters and enhancers to increase gene expression in the target tissue.

However, like all gene therapy vectors, pDNA has its drawbacks. One of the biggest concerns is its potential for immunogenicity, which can cause an immune response and limit the effectiveness of the therapy. Therefore, careful consideration must be given to the design and delivery of pDNA to minimize this risk [36,37,38,39,40,41,42,43,44,45].

### 2.9. Messenger Ribonucleic Acid (mRNA)

mRNA is a promising drug for gene therapy. It is a single-stranded molecule of RNA that corresponds to the genetic sequence of a gene and is read by a ribosome in the process of protein synthesis. By delivering corrected mRNA into cells, they can receive the right blueprint for creating healthy proteins, which can help treat a variety of genetic disorders.

One of the major advantages of mRNA as a drug is its ease of manipulation. It can be rapidly produced and modified to fit specific needs. Additionally, it offers transient expression, meaning protein production is not permanent and can be turned off if necessary. Moreover, mRNA is adaptive and can be converted without mutagenesis.

However, there are some downsides to using mRNA for gene therapy. One of the major drawbacks is its immunogenicity, meaning the body’s immune system may recognize it as foreign and attack it, leading to negative side effects. Additionally, it can be challenging to control the concentration of reporter mRNA, which can lead to unintended effects [46,47,48].

### 2.10. Meganucleases

Meganucleases are a promising tool for targeted gene editing, characterized by a large recognition site and high specificity. They can be thought of as molecular DNA scissors, capable of replacing, eliminating, or modifying specific sequences in a highly precise manner.

The small size of meganucleases allows for their use with many viral vectors, and they can tolerate some mismatches, resulting in low off-target editing.

However, the design and reengineering of meganucleases for new specificities can be extremely challenging.

Despite these obstacles, recent research has demonstrated the potential of meganucleases to induce homologous recombination in yeast and mammalian cells, highlighting their potential as a tool for precise gene editing in various applications [49,50,51,52,53].

### 2.11. Zinc Finger Nucleases (ZFNs)

ZFNs are artificial endonucleases that have been developed for targeted gene editing. They are composed of a designed ZFP and the cleavage domain of the FokI restriction enzyme. The ZFP is engineered to recognize and bind to specific DNA sequences, while the FokI domain cleaves the DNA at the target site.

One advantage of ZFNs is that they can tolerate some mismatches, which reduces off-target editing. However, G-rich regions of DNA can be challenging for ZFNs, and their design can be difficult. Multiplexing, or targeting multiple genes at once, is also a challenge with ZFNs [54,55].

### 2.12. Transcription Activator-like Effector Nucleases (TALENs)

TALENs are a promising gene therapy tool that can be used to cut specific sequences of DNA. TALENs are restriction enzymes that have been engineered to bind and cut DNA in a highly specific manner.

Their size ranges from 32 to 40 base pairs, and they are capable of tolerating some mismatches, resulting in low off-target editing. Additionally, TALENs have moderate design requirements.

However, TALENs do have some limitations. Specifically, they require a 5’ T for each TALEN and are challenging to multiplex. Furthermore, their large size makes it difficult to utilize viral vectors, and repetitive sequences can lead to unwanted recombination [56,57,58,59,60].

### 2.13. DNA Aptamers

DNA aptamers are short sequences of artificial DNA that can bind specific target molecules with extremely high affinity based on their structural conformations. These aptamers are becoming increasingly popular in various biosensing and therapeutic applications due to their stability and the ease of their generation and synthesis. They are also known for having almost no immunogenicity and for their efficient penetration, lower batch variation, easy modification, cost-effectiveness, and short production times.

DNA aptamers have been compared with other biorecognition elements, such as antibody scFv and antibody Fab’ fragments. They have also been used to select molecules that bind to specific targets, such as gluten, cocaine, and malachite green. In addition, their secondary structural requirements have been investigated through thermodynamic and mutation studies. The 2.8 A crystal structure of the malachite green aptamer and the structural investigations of RNA and DNA aptamers in solution have been described in detail in previous studies [61,62,63,64,65,66,67,68,69,70].

### 2.14. Tumor Necrosis Factor-Related Apoptosis-Inducing Ligand (TRAIL/APO2L)

TRAIL is a member of the tumor necrosis factor family. Its main function is to induce apoptosis, or programmed cell death, in cancerous cells by binding to the death receptor 4 (DR4) or DR5. Unlike other treatments, TRAIL has the advantage of selectively targeting tumor cells while avoiding side effects in normal tissues. This makes TRAIL a promising therapy for cancer treatment.

One of the most significant benefits of TRAIL is its ability to efficiently kill tumor cells. By activating the apoptotic pathway in cancer cells, TRAIL induces their death, halting their growth and spread. However, some tumor cells are resistant to TRAIL, which limits its effectiveness. The existence of TRAIL-resistant tumor cells remains a challenge that needs to be overcome to maximize the benefits of TRAIL therapy.

Despite the limitations, TRAIL is a promising therapy for cancer treatment due to its selective toxicity towards cancer cells. Researchers are working to overcome the problem of TRAIL resistance by combining it with other treatments such as chemotherapy or radiation therapy. These combinations have shown promising results in clinical trials and may improve the effectiveness of TRAIL therapy in the future.

In conclusion, TRAIL has emerged as a promising therapy for cancer treatment due to its ability to selectively target tumor cells. While the existence of TRAIL-resistant tumor cells limits its effectiveness, ongoing research aims to overcome this challenge and improve the therapeutic potential of TRAIL [71,72].

### 2.15. Phosphorodiamidate Morpholino Oligomers (PMOs)

PMOs are short, single-stranded DNA analogs built upon a backbone of morpholine rings connected by phosphorodiamidate linkages. These uncharged nucleic acid analogs are designed to bind to complementary sequences of target mRNA through Watson–Crick base pairing, which results in the blocking of protein translation through steric blockade.

PMO oligomers range in size from 6 to 22 base pairs and have been shown to be resistant to a variety of enzymes present in biologic fluids, making them ideal for in vivo applications.

The resistance of PMO to nucleases and other enzymes is a major advantage for their use in gene therapy. By preventing degradation, they can efficiently target and inhibit translation of specific mRNA molecules. PMOs have been used in clinical trials to treat various diseases such as Duchenne muscular dystrophy and spinal muscular atrophy, demonstrating their therapeutic potential [73,74].

### 2.16. Naked DNA

Naked DNA, which is simply DNA without any associated proteins, has been widely investigated as a gene transfer tool for several tissues including skin, thymus, cardiac muscle, skeletal muscle, and liver cells. This method involves direct injection of DNA into the target tissue, allowing for the transfer of a gene with a size range of 2–19 kb.

Naked DNA-based gene transfer is a safe and straightforward approach, but its application is limited to certain areas such as DNA vaccination. In skeletal muscle, long-term expression has been observed after injection for more than 19 months.

Despite these advantages, the efficiency of naked DNA for gene delivery is low, with less than 1% of total myofibers showing transgenic expression following a single injection. However, repeated injections can improve the overall transfection efficiency, making naked DNA a viable option for certain gene therapy applications [75].

## 3. Forty-Six Gene Therapy Carriers

Within this section, and subsequently in Table 2, our focal point shall rest upon the second facet of gene therapy, namely carriers, which serve as conduits for the delivery of gene therapy drugs to their designated destinations. We aim to proffer an exhaustive compendium of carriers presently employed within gene therapy practices, accompanied by a thorough discourse on their respective constraints, benefits, drawbacks, and authorized utility within gene therapy products, as well as illustrative instances highlighting their applications.

### 3.1. Viral Vectors

As nature’s original “gene therapists”, viruses possess the necessary tools for efficient cellular entry and dissemination of their genetic payloads. Scientists have taken advantage of this natural capability and repurposed viruses for human gene therapy by replacing their original genetic material with therapeutic gene therapy drugs. This process has led to the development of numerous viral vectors that are now used in gene therapy. In the following sections, we will explore some of the most widely used viral vectors for gene therapy.

#### 3.1.1. Retroviral

Ex vivo gene therapy is a process of genetically modifying cells outside of the body and then returning them back to the patient. Retroviral vectors are one of the commonly used viral vectors for ex vivo gene therapy, and they have been used in various clinical trials. Retroviruses are small RNA viruses that replicate through a DNA intermediate. These vectors are suitable for ex vivo gene therapy, such as transducing CD34+ bone marrow hematopoietic stem cells or peripheral blood lymphocytes. The retroviral vector has a size limit of 10 kb, which means that it can deliver genetic material of a limited size.

However, retroviral vectors have some limitations. They have a low vector titer, a low in vitro transfection efficiency, particle instability, and difficulty in concentrating. Moreover, they cannot transduce non-dividing post-mitotic cells, and their particles can only infect proliferating cells. This can limit the therapeutic application of these vectors.

Despite these limitations, retroviral vectors have been used in several approved gene therapies. Invossa (TissueGene-C or Tonogenchoncel-L), YESCARTA (axicabtagene ciloleucel), Zalmoxis (Allogenic T cells encoding LNGFR and HSV-TK), Strimvelis (GSK-2696273), and TECARTUS (brexucabtagene autoleucel) are examples of retroviral vector-based approved products. The targets of these products include growth factor β1 (TGF β1), ΔLNGFR and HSV-TK Mut2, ADA cDNA, Anti-CD19 CAR gene, and Cyclin G1.

In summary, retroviral vectors have shown their potential in ex vivo gene therapy, especially for transducing hematopoietic stem cells or peripheral blood lymphocytes. However, a low efficiency and limitations of retroviral vectors have also been reported [75,79,80].

#### 3.1.2. Lentiviral Vector (LVV)

LVVs are a type of viral vector that is spherical and composed of single-stranded RNA. These vectors are commonly used in ex vivo gene therapy applications for T cells. The LVVs have a higher limit of 8 kb, making them suitable for larger genes. One of the key advantages of LVVs is their ability to persistently transfer genes in dividing cells. However, they may also pose a safety risk as they can lead to unnecessary non-target insertion mutations.

LVVs have the ability to transduce both dividing and non-dividing cells, which can be beneficial in some applications but may lead to oncogenesis in others. Despite the potential risks, there are several approved gene therapy products that utilize LVVs. These include ABECMA (idecabtagene vicleucel), CARVYKTI (ciltacabtagene autoleucel), SKYSONA (elivaldogene autotemcel), Kymriah (tisagenlecleucel CTL 019), BREYANZI, and ZYNTEGLO (betibeglogene autotemcel). The approved products use LVVs to deliver genes such as a chimeric antigen receptor (CAR), anti-B cell maturation antigen (BCMA), ABCD1 gene, and Beta-A-T87Q-globin [75,80].

#### 3.1.3. Adeno-Associated Virus (AAV)

Adeno-associated virus (AAV) is a non-enveloped, small, double-stranded DNA virus that has gained popularity as a vector for gene therapy. AAV is a good choice for in vivo gene therapy as it has good biological characteristics and is non-immunogenic and non-pathogenic. AAV has a high gene transduction efficiency and large-scale ease of use, which has led to its use in a range of somatic gene therapy applications.

One of the limitations of AAV is its packaging capacity, which is limited to 4.5 kb. However, AAV is known for its genetic stability, making it an attractive vector for gene therapy. The production process for AAV is complicated and expensive, which may be a barrier to its wider use.

Despite these challenges, there have been several approved products that use AAV as the vector for somatic gene therapy, including LUXTURNA and Zolgensma. LUXTURNA targets the RPE65 gene to treat inherited retinal dystrophy, while Zolgensma targets the SMN1 gene in motor neurons to treat spinal muscular atrophy [75,80,81].

#### 3.1.4. Adenovirus (AdV)

Adenovirus (AdV) is an in vivo somatic viral vector with an icosahedral nucleocapsid structure.

It is easy to purify and has genetic stability and a large foreign gene loading capacity of up to 48 kb, making it efficient for transduction in most tissues.

Gendicine and Oncorine are two approved AdV-based gene therapies targeting the Tp53 tumor-suppressor gene.

However, AdV may initiate a strong inflammatory response, which is a major limitation of its use [7].

#### 3.1.5. Herpes Simplex Virus Type 1 (HSV-1)

Herpes simplex virus type 1 (HSV-1) is a viral vector that has shown great promise for somatic gene therapy in vivo. With a packaging capacity of up to 40 kb (replication defective) and 150 kb (amplicon), HSV-1 can accommodate large transgene cassettes, making it an attractive option for gene therapy. In addition, the strong tropism of HSV-1 for neurons makes it an ideal candidate for therapy of central nervous system diseases.

However, the use of HSV-1 as a vector has some limitations. It may initiate a strong inflammatory response, which can be a concern for safety in some cases. Transient gene expression in cells other than neurons is also a limitation. Despite these limitations, the use of HSV-1 as a vector has been approved for certain products, including IMLYGIC (Talimogenelaherparepvec).

IMLYGIC is a modified HSV-1 vector encoding the human granulocyte-macrophage colony-stimulating factor (GM-CSF) gene, which has been approved for the treatment of advanced melanoma [80,82].

#### 3.1.6. Alphavirus

Alphaviruses are single-stranded RNA viruses that have been used in gene therapy as viral vectors for in vivo somatic cell delivery. These enveloped alphavirus particles are made up of a protein capsid structure surrounded by spike membrane proteins. They recognize surface proteins such as laminin and heparin receptors on mammalian and insect cells, delivering the RNA genome to the cell cytoplasm for immediate RNA replication.

One of the advantages of using alphavirus vectors is their broad host range and high level of transient heterologous gene expression. However, the production of expensive virus stocks and the highly transient nature of heterologous gene expression are some of the disadvantages. The limit of alphavirus vectors is less than 12 kb [83,84].

#### 3.1.7. Poxvirus

Poxviruses are a class of double-stranded DNA viruses that belong to the family Poxviridae.

They have been used as viral vectors for gene therapy due to their ability to induce long-lasting immunological responses. In nature, vertebrates and arthropods serve as natural hosts for this virus. With 83 species currently identified, divided among 22 genera and two subfamilies, poxviruses have shown great potential as oncolytic therapeutics for various cancer types, with some successes observed in phase I/II clinical trials. The maximum cargo capacity of poxviral vectors is limited to 500–1000 bp.

However, the therapeutic potential of most poxviruses has yet to be fully characterized and developed [85,86,87,88].

#### 3.1.8. Bacteriophage

Bacteriophages, also known as phages, are a type of viral vector that have become increasingly popular in gene therapy. These viruses infect and replicate only in bacterial cells and are the most abundant biological agent on Earth. They consist of a nucleic acid genome encased in a shell of phage-encoded capsid proteins.

Bacteriophages are highly specific, making them ideal for targeted therapy. They are also easy and inexpensive to propagate, highly stable, and can be used to package non-phage nucleic acid, proteins, or other types of materials. Additionally, they are easy to modify and are GRAS agents.

There are various types of bacteriophages with different genome sizes, including 5 Kbp, 49 Kbp, 40 Kbp, and 160 Kbp–250 Kbp. Their small size makes them useful for gene therapy applications that require small vectors [89,90,91,92,93,94,95].

#### 3.1.9. Epstein–Barr Virus (EBV), Human Gammaherpesvirus 4

The Epstein–Barr virus (EBV), also known as human gammaherpesvirus 4, is a double-stranded DNA virus with a limit of 172 Kbp.

One of the advantages of using the EBV vector is that it provides both long-term and timely controlled exogenous gene expression in neurons. Additionally, the vector remains in the nucleus as an episome and does not integrate into the host genome. This feature helps reduce the risk of insertional mutagenesis, a common concern in gene therapy [96,97,98,99,100].

### 3.2. Non-Viral Vectors

The development of non-viral vectors for gene therapy has been a major focus in recent years in order to address the limitations of viral vectors, such as their high cost of production, complex manufacturing processes, potential for inducing inflammatory responses, etc. Non-viral vectors offer a more affordable, simpler, and effective alternative. Numerous non-viral vectors have been created, each with unique advantages and disadvantages. In the following section, we will explore some of the most notable non-viral vectors used in gene therapy.

#### 3.2.1. Lipid Nanoparticles (LNPs)

Lipid nanoparticles (LNPs) are a type of non-viral vector that are formed by the orientation of phospholipid biomolecules. These biomolecules can encapsulate both fat-soluble and water-soluble drugs, and deliver genetic drugs to the body through fusion with cell membranes. LNPs have a limit of approximately 22 base pairs and have several advantages such as good biocompatibility, reduced drug toxicity, drug resistance, and endosomal escape facilitation. These characteristics make LNPs an attractive option for gene therapy [75,101].

#### 3.2.2. Amphiphilic Phospholipid Peptide Dendrimers (AmPPDs)

Non-viral vectors, such as amphiphilic phospholipid peptide dendrimers (AmPPDs), have been extensively studied in recent years due to their potential as effective siRNA delivery systems for anticancer therapy. AmPPDs are capable of compacting siRNA into nanoparticles to protect it from enzymatic degradation. This vector bears the natural lipid derivative DSPE as the hydrophobic tail and different dendritic l-lysine as the hydrophilic head, with a limit of about 13 kDa.

One of the advantages of AmPPDs is their simplicity of synthesis and versatility of functionalization. However, they also have some limitations.

One drawback is their relatively low loading capacity for siRNA, which may limit their efficacy in some applications. Another potential issue is the premature leakage of payloads, which can lead to non-specific effects and potential toxicity [102].

#### 3.2.3. Gold Nanoparticles (AuNPs)

Gold nanoparticles (AuNPs) have become a promising non-viral vector in gene therapy, with advantages including facile surface modification and unique optical properties. These inorganic biomaterials can be synthesized and modified with chemical and biological molecules, making them a versatile tool for drug delivery and molecular diagnostics.

The surface of AuNPs supports the efficient attachment of various biomacromolecules via chemisorption, chemical conjugation, and electrostatic interactions. In addition, their small size of 1–100 nm and non-toxicity make them an attractive option for use in gene therapy.

AuNPs have been extensively used as drug carriers for the intracellular delivery of therapeutics and as molecular nanoprobes for the detection and monitoring of target molecules of interest [103,104,105,106,107,108].

#### 3.2.4. REDV Peptide-Modified TMC-g-PEG Polyplex

The REDV peptide-modified TMC-g-PEG polyplex is designed to target vascular endothelial cells (VECs) and deliver miRNA-126 to these cells. The vector is composed of a short peptide, Arg-Glu-Asp-Val (REDV), linked to trimethyl chitosan (TMC) via a bifunctional poly(ethylene glycol) (PEG) linker.

This complex has potential as a miRNA carrier for rapid endothelialization in artificial blood vessels [109].

#### 3.2.5. PEGylated Nanoparticles

PEGylated nanoparticles are a non-viral vector, formed by coating the surface of nanoparticles with polyethylene glycol (PEG), a process known as “PEGylation”. PEGylation has several advantages over traditional viral vectors, including an improved systemic circulation time and decreased immunogenicity.

By shielding the surface of the nanoparticles from aggregation, opsonization, and phagocytosis, PEGylation can prolong the systemic circulation time, allowing for more efficient delivery of therapeutic genes. This technology has shown great promise in preclinical studies and is currently being tested in clinical trials for a variety of diseases, including cancer, cystic fibrosis, and cardiovascular disease [110].

#### 3.2.6. PBAT (Polybutylene Adipate Terephthalate) Polymer

PBAT (polybutylene adipate terephthalate) polymer is a biodegradable random copolymer consisting of adipic acid, 1,4-butanediol, and terephthalic acid, and has shown great potential for both in vitro and in vivo gene delivery.

One of the major benefits of PBAT is its biodegradability. Unlike other non-viral vectors, PBAT is fully biodegradable, making it an attractive option for gene therapy applications. In addition, PBAT has demonstrated a high transfection efficiency and low cytotoxicity when delivering a variety of genetic material, including DNA, mRNA, and Cas9 RNP.

PBAT’s ability to deliver genetic material with high transfection efficiency and low cytotoxicity is attributed to its unique structure. The random copolymer has a hydrophobic core and a hydrophilic shell, which allows it to encapsulate the genetic material and protect it from degradation while also promoting cell uptake [111,112,113,114,115,116].

#### 3.2.7. GSH-Responsive Polyplexes

GSH-responsive polyplexes are an exciting non-viral vector for gene therapy that show great promise in delivering genetic biomacromolecules. These polyplexes are capable of forming electrostatic interactions with negatively charged DNA, mRNA, and Cas9/sgRNA ribonucleoprotein (RNP), allowing for efficient cellular uptake, endosomal escape, and cytosol unpacking of payloads. Furthermore, this delivery system is universal and glutathione-responsive, making it highly adaptable for a wide range of therapeutic applications.

In vitro studies have shown a relatively high transfection efficiency with low cytotoxicity, making GSH-responsive polyplexes an excellent alternative to viral vectors. Additionally, the convenient surface functionalization of these polyplexes makes them easy to modify for specific applications [117].

#### 3.2.8. Magnetic Nanoparticles

Magnetic nanoparticles, a type of non-viral vector, have garnered considerable interest in recent years for their potential in gene delivery. These nanoparticles consist of a magnetic material, typically iron, nickel, or cobalt, and a functional chemical component. Magnetic nanoparticles are intrinsically biocompatible, and their magnetic moments can be controlled using externally applied magnetic fields to leverage their nanoscale behavior. Additionally, magnetic nanoparticles are relatively easy to synthesize and modify, making them an attractive option for gene therapy applications.

One potential application of magnetic nanoparticles in gene therapy is their use in targeted delivery to specific cells or tissues. By functionalizing the surface of magnetic nanoparticles with targeting moieties, such as antibodies or peptides, these particles can be directed to specific cell types or even individual cells within a tissue. This can increase the specificity and efficacy of gene therapy while minimizing off-target effects.

Another potential advantage of magnetic nanoparticles is their ability to penetrate barriers such as the blood–brain barrier, allowing for non-invasive gene delivery to the central nervous system. However, further research is needed to fully understand the potential of magnetic nanoparticles in gene therapy and to optimize their delivery efficiency and safety [118,119,120,121,122,123,124].

#### 3.2.9. Ethylcellulose Nano-Emulsions

Ethylcellulose nano-emulsions can be prepared using a low-energy approach with aqueous components, resulting in folate–ethylcellulose nanoparticle complexes.

Ethylcellulose is an abundant and biocompatible material, listed by the Food and Drug Administration as “generally recognized as safe” (GRAS). The surface charge of these nanoparticles can be adjusted to be either positive or negative, allowing for optimal interaction with the desired genetic biomacromolecule. This versatility makes ethylcellulose nano-emulsions an attractive candidate for gene delivery in various applications [125].

#### 3.2.10. Lipoplexes

Lipoplexes are a class of non-viral vectors used in gene therapy that have attracted attention due to their ability to effectively deliver genetic material into target cells. These vectors consist of cationic lipids that are amphiphilic in nature, meaning they have both hydrophilic and hydrophobic regions. Typically, a charged cationic headgroup is attached via a linker, such as glycerol, to a double hydrocarbon chain or cholesterol derivative. Lipoplexes can be used both in vitro and in vivo for gene delivery applications.

One of the main advantages of lipoplexes is that they are relatively easy to produce compared to viral vectors. In addition, they do not induce a host inflammatory or immune response, which can be a significant advantage for gene therapy applications.

Despite these benefits, there are still challenges associated with the use of lipoplexes, including concerns about their stability and efficiency in delivering genetic material to target cells [126,127].

#### 3.2.11. Polyplexes

Non-viral vectors such as polyplexes can overcome some of the limitations of viral vectors, such as safety concerns and host immune responses. Polyplexes consist of cationic polymers and genes, which form nanoscale complexes through electrostatic interactions.

These complexes have shown high versatility, low toxicity, low immunogenicity, and biodegradability. Furthermore, polyplexes are able to form complexes with small RNAs, leading to RNA protection, cellular delivery, and intracellular release [128,129].

#### 3.2.12. Micelleplexes

Micelleplexes, denoting nanostructured NA/micelle-like complexes, are characterized by their distinctive attributes forged by the chemical composition of amphiphilic copolymers. These copolymers showcase regions that are both cationic and hydrophilic/hydrophobic, thereby facilitating the compaction of nucleic acids through interaction with one or more cationic constituents.

The virtues of micelleplexes extend to proficient binding, conveyance, and precision delivery of nucleic acids to neoplastic cells. These micelleplexes notably outperform their polyplex counterparts in the domains of gene silencing, cellular internalization, toxicity mitigation, colloidal stability enhancement, and payload trafficking efficiency.

Nevertheless, the progressive landscape of innovation is attended by a dualistic complexity, where nanosystems characterized by an excessive positive charge density beckon forth discerning considerations concerning their in vivo toxicity potential, thereby warranting scrupulous evaluation [130,131].

#### 3.2.13. Carbon Nanotubes (CNTs)

Carbon nanotubes (CNTs) are non-viral vectors with unique physical and chemical properties. These allotropic forms of carbon are cylinder-shaped and can be produced through chemical vapor deposition. CNTs have a high surface area, a needle-like structure, considerable strength, flexible interaction with cargo, high drug loading capacity, and outstanding optical and electrical features, making them promising for targeted gene delivery. Additionally, CNTs have high stability, biocompatibility, and the ability to release therapeutic agents at targeted sites.

However, one important drawback is the concern over nanotoxicology. Further studies are necessary to fully understand the potential risks and benefits of using CNTs as a gene delivery system [132].

#### 3.2.14. Graphene

Graphene is a non-viral vector that has been widely studied for its potential use in gene therapy. As a single layer of carbon atoms arranged in a hexagonal lattice, graphene is incredibly thin, lightweight, and strong. However, its use as a gene therapy vector is not without challenges.

One major concern is its potential toxicity, as well as the risk of inappropriate release of therapeutic agents.

Despite these challenges, the unique properties of graphene have led researchers to explore its use as a gene delivery vehicle. Its high surface area, flexibility, and biocompatibility make it an attractive option for delivering therapeutic agents to target cells. Additionally, its ability to penetrate cell membranes and interact with DNA makes it a promising tool for gene therapy [133,134].

#### 3.2.15. GO-PEI-10k Complex

The GO-PEI-10k complex has gained attention for its low cytotoxicity and high transfection efficiency. This complex is formed by binding GO with a cationic polymer, polyethyleneimine (PEI), with two different molecular weights of 1.2 kDa and 10 kDa. The resulting complex is stable in physiological solutions and has been shown to exhibit a significantly reduced toxicity to treated cells compared to the bare PEI-10k polymer.

Further studies have demonstrated the potential of GO–PEI complexes to bind with pDNA for intracellular transfection of genes. In particular, the GO-PEI-10k complex has been shown to efficiently transfect the enhanced green fluorescence protein (EGFP) gene in HeLa cells [134].

#### 3.2.16. Quantum Dots (QDs)

Quantum dots (QDs) have been proposed as an innovative type of non-viral vector in gene therapy. These semiconductor nanocrystals have unique electronic and optical properties that are different to the bulk material thanks to quantum mechanics. QDs have a narrow emission peak, a size-dependent emission wavelength, and a broad excitation range that could be exploited for several biomedical applications such as molecular imaging, biosensing, and diagnostic systems.

One significant advantage of QDs is that they are biocompatible, stable, and soluble in the biomatrix. However, like many other nanomaterials, QDs also have some potential toxicity concerns that should be taken into account [135,136].

#### 3.2.17. Graphene Quantum Dots (GQDs)

Graphene quantum dots (GQDs) are nanoparticles of graphene with sizes smaller than 100 nm that exhibit low toxicity, stable photoluminescence, and excellent chemical and thermal stabilities. They also have a pronounced quantum confinement effect, which makes them suitable for biological, opto-electronic, energy, and environmental applications.

As a non-viral vector, GQDs have been found to have dual functions for gene delivery and nuclear targeting. They are nontoxic carriers that can effectively deliver therapeutic genes to the target cells, making them an attractive option for gene therapy. Additionally, GQDs can be modified to enhance their stability and biocompatibility, further increasing their potential as a gene delivery vector [137,138,139,140,141,142].

#### 3.2.18. Up-Conversion Nanoparticles (UCNs)

These optical nanocrystals are doped with lanthanide ions and can convert low-energy near-infrared (NIR) light into high-energy visible or ultraviolet light, which makes them a photoactive delivery platform for gene carriers. UCNs have unique properties such as a high quantum yield, high stability, and low photobleaching, which make them attractive for biomedical applications. Additionally, surface modifications can be made to improve UCNP-based gene carriers, allowing for an improved biological efficiency [143].

#### 3.2.19. Mesoporous Silica Nanoparticles (MSNs)

Mesoporous silica nanoparticles (MSNs) are an attractive vector for gene therapy due to their tunable size, biocompatibility, and high surface area. The porous structure of MSNs provides a large surface area, allowing functional groups to be attached to the particle surface. MSNs offer easily tunable particle and pore sizes, greater surface areas, and a simple mesoporous or hollow structure, making them ideal for drug delivery applications.

Additionally, MSNs are stimuli-responsive, with pH-sensitive and redox-sensitive drug release profiles. Although MSNs have been mainly tested in vitro, recent studies have also demonstrated their efficacy in vivo [144,145,146].

#### 3.2.20. Ferritin

Ferritin, a spherical nanocage formed by the self-assembly of heavy and light polypeptide chains, is a promising non-viral vector for gene therapy due to its natural transport function for metals and versatility in cargo loading. Ferritin has a small and uniform size, limited to 8 nm in diameter, which allows for efficient cellular uptake and intracellular trafficking. It is also highly stable under various conditions, including high pHs and temperatures.

Additionally, ferritin can be visualized using magnetic resonance imaging (MRI) for in vivo tracking purposes.

However, it is important to note that ferritin can elicit robust humoral immunogenicity, which may limit its application in certain gene therapy contexts [147,148,149,150].

#### 3.2.21. Red Blood Cell Membrane (RBCM)

The Red Blood Cell Membrane (RBCM) is a non-viral vector that has gained attention for its biocompatibility, biodegradability, and long circulating half-life. Erythrocytes, or RBCs, are the most abundant circulating cells in the blood, and have been widely used in drug delivery systems (DDSs). By using a “camouflage” consisting of erythrocyte membranes, nanoparticles can imitate RBCs and achieve long-term circulation in the blood of animal models. This approach combines the advantages of native erythrocyte membranes with those of nanomaterials.

Coating nanoparticles with RBC membranes has several advantages. Firstly, they are able to escape the immune system and achieve long-term circulation in the body. Secondly, they have inherent biocompatibility and biodegradability and can avoid some of the toxicities associated with other nano-preparations. Additionally, RBCs have a lifespan of up to 120 days, allowing for sustained release of cargo. Furthermore, the large quantities of cell membranes make it easy to achieve a high load capacity, while also improving nanoparticle stability, enhancing in vitro storage time, and discouraging aggregation [151].

#### 3.2.22. Gelatin Nanoparticles

Gelatin nanoparticles have shown great potential due to their excellent immune evasion capabilities. One of the advantages of these nanoparticles is their susceptibility to hydrolysis by broad-spectrum bacterial gelatinases, which can convert them into small biomolecules. This property can be advantageous in cases where the nanoparticles need to be rapidly degraded and cleared from the body. Furthermore, the biodegradable and biocompatible nature of gelatin makes it an attractive choice for therapeutic applications [151].

#### 3.2.23. DNA Nanoclews (NCs)

DNA nanoclews (NCs) are made using rolling circle amplification (RCA) and have a uniform size, high biodegradability, and spatial addressability. They are particularly effective at escaping endosomes due to the use of PEI, which coats the Cas9/sgRNA/NC complex.

While DNA NCs are still in the early stages of development, studies have shown that they can be used in both in vitro and in vivo settings. In fact, these vectors have been used to treat diseases such as cystic fibrosis and HIV. In addition, they are easy to synthesize and can be modified to target specific cells or tissues.

One of the key advantages of DNA NCs is their uniform size, which allows for better control over their delivery and release. This is important because it can reduce toxicity and increase the efficiency of the treatment. Additionally, DNA NCs have been shown to be highly biodegradable, meaning they are easily broken down by the body and do not accumulate in tissues.

Another important advantage of DNA NCs is their spatial addressability, which refers to their ability to be modified with various functional groups. This allows them to be targeted to specific cells or tissues, which can improve the efficiency and effectiveness of the therapy. Finally, DNA NCs have been shown to be effective at escaping endosomes, which is a critical step in delivering the therapeutic payload to the target cell [152,153,154].

#### 3.2.24. Arrestin Domain-Containing Protein 1 (ARRDC1)-Mediated Microvesicles (ARMMs)

Arrestin domain-containing protein 1 (ARRDC1)-mediated microvesicles (ARMMs) are capable of delivering various macromolecules, including the tumor suppressor p53 protein, RNAs, and the genome-editing CRISPR-Cas9/guide RNA complex, both in vitro and in vivo. ARMMs are particularly attractive for gene therapy due to their lipid bilayer membrane which protects the enclosed cargo from degradation by proteases or nucleases and also shields them from being recognized as foreign antigens by the immune system. ARMMs have demonstrated high efficiency in targeted gene delivery with minimal off-target effects.

ARMMs are produced through the ARRD1 protein, which packages the desired cargo into small vesicles that are transported out of the cell. These vesicles can then be taken up by other cells where the contents can be released to perform their therapeutic function.

One of the major advantages of using ARMMs for gene therapy is their ability to target specific tissues or organs. This feature could potentially reduce side effects associated with current gene therapy approaches [155,156].

#### 3.2.25. Multifunctional Peptides

Multifunctional peptides are able to serve as building blocks for self-assembling nanoscale structures. Peptides can be rationally designed to target molecular recognition sites, and their biocompatibility and biodegradability make them an attractive option. Moreover, peptides offer the possibility of endless combinations and modifications of amino acid residues, allowing for the assembly of modular, multiplexed delivery systems [157].

#### 3.2.26. Cell-Penetrating Peptide (CPP), Cell-Permeable Peptides, Protein Transduction Domains (PTDs)

Cell-penetrating peptide (CPP)-modified carriers are short peptides that facilitate cellular uptake of various molecular cargo, including DNA, by crossing cell membranes. These carriers typically consist of small peptide domains of less than 40 amino acids, which have the ability to penetrate cells easily. CPPs recognize specific types of cells and have been shown to translocate across their cellular membranes.

One of the major advantages of CPP-modified carriers is their specificity in targeting certain types of cells. This specificity is derived from the amino acid sequence of the CPP, which determines its binding and internalization properties. In addition, CPPs can translocate across cellular membranes without the need for any additional endosomal escape mechanism, which can further enhance gene delivery efficiency.

CPP-modified carriers have been used for both in vitro and in vivo applications in gene therapy. They have been successfully employed to deliver various types of genetic material, such as DNA, RNA, and proteins, to a wide range of cell types. These carriers can also protect the genetic material from degradation and clearance, leading to sustained gene expression [158,159].

#### 3.2.27. Exosomes (Endogenous Nanocarriers)

Exosomes are secreted nanoparticles that can be defined by their size, surface protein, and lipid composition. They are produced by many types of cells and can be harvested from various bodily fluids. One of the advantages of exosomes is that they can be tailored for specific therapeutic purposes. Exosomes can be derived from different cell types, such as immune cells, stem cells, and cancer cells, to achieve specific effects.

Additionally, exosomes can be loaded with various therapeutic cargoes, including small molecules, RNA, and proteins. Targeting peptides can also be added to exosomes to enhance their specificity for particular cells or tissues. Moreover, the method of loading and the administration route can be optimized to improve the efficiency and specificity of exosome-mediated delivery [160].

### 3.3. Physical Delivery

In the world of gene therapy, in addition to the therapeutic genes and their carriers, there are also physical delivery methods that can enhance the efficiency and effectiveness of gene delivery. We can think of these physical vectors as accelerators that improve the performance of the gene therapy vehicles. Some of these physical vectors can even act as gene delivery vehicles themselves. Let us explore the various physical vectors used in gene therapy to learn more about their unique characteristics and advantages.

#### 3.3.1. Microneedles (MNs)

Microneedles (MNs) are a promising tool in gene therapy that can penetrate the stratum corneum, which is the main barrier for drug delivery through the skin. There are different types of MNs, such as metal MNs, coated MNs, and dissolving MNs. MNs have been shown to be effective in delivering both low-molecular-weight and high-molecular-weight agents, including nucleic acids, with ease of administration and without significant pain. This pain-free and patient-friendly feature of MNs lends it the potential for self-administration. Moreover, the low production cost of MNs makes it an attractive option for marketing. MNs have been used for the transdermal delivery of siRNA, low-molecular-weight drugs, oligonucleotides, DNA, peptides, proteins, and inactivated viruses.

However, MNs do have some limitations. The application site is limited to areas such as the arms, hands, and abdomen. Additionally, the duration time for application is critical since the microneedles dissolve completely in 20 min. Although MNs may not be better than intradermal injection alone, the combination of MNs and electroporation has been found to be more effective [161].

#### 3.3.2. Gene Gun (Biolistic Particle Delivery System)

The gene gun, also known as a biolistic particle delivery system, is a technique used to incorporate DNA or RNA into cells that are typically difficult to transfect using traditional methods. This approach involves using microparticles, tiny projectiles with a diameter of about 1 μm, which are propelled into tissues by pressurized gas. This method is gaining popularity due to its high efficiency in delivering DNA into cells.

However, there is a possibility of significant tissue damage during the process. Despite these limitations, biolistic transfection has shown promising results in inducing immune responses against infectious diseases and cancer [162].

#### 3.3.3. Jet Injection

Jet injection is a physical delivery method that uses high pressure to force microdroplets of liquid to penetrate the skin to deliver a drug or vaccine into intradermal, subcutaneous, or intramuscular tissues. Unlike needle-based injections, jet injection minimizes patient discomfort and eliminates the risk of needlestick injury. A high-pressure liquid stream is used to pierce the skin or target tissue, with the pressure reaching up to 3–4 bar and the velocity of the droplets ranging from 100 ms^−1^ to 200 ms^−1^.

Jet injection has been used in DNA vaccination to induce the host’s immune response to the gene product encoded and expressed by the plasmid used. However, it has been noted that the method may cause tissue damage due to the high-pressure stream and it may not be suitable for delivering molecules with certain physical properties, such as a high viscosity or a high molecular weight.

Overall, jet injection remains a promising method of gene delivery that offers improved patient comfort and reduced risk of needlestick injury. Further research is needed to optimize the technique for specific molecules and tissue types [163].

#### 3.3.4. Electroporation

Electroporation has emerged as a widely used method for efficient gene delivery in gene therapy. It involves the use of short high-voltage pulses to temporarily break down the cell membrane, allowing for the transfer of genetic material into cells.

The advantages of electroporation include a significant increase in gene delivery and expression, as well as a decrease in development costs and time. It is also safe and highly effective in delivering larger and more complex payloads, such as CRISPR/Cas9 systems, DNA vectors, and even small molecules, to various tissues, including the muscles, skin, heart, liver, lungs, and vasculature.

However, optimization is required for each new setting, as there can be variable transfection efficiency, limited cell viability, and potential tissue damage [164,165,166,167,168].

#### 3.3.5. Ultrasound

Ultrasound, a noninvasive technique, has been explored as a means of delivering genes into cells in a process known as sonoporation. The ultrasound waves cause cavitation, resulting in the formation of pores in the cell membrane that allow the entry of therapeutic DNA into cells. This method of gene transfer is highly efficient and has a more favorable safety profile than other non-viral delivery methods. In addition, ultrasound-mediated gene transfer is considered to be less invasive than electroporation and is more readily accepted in clinical settings.

However, careful control of ultrasound parameters is required to avoid potential side effects. Despite this limitation, ultrasound-mediated gene transfer has proven to be an efficient and effective technique, with systemic injection of DNA possible. Researchers have found that sonoporation can significantly increase the uptake and expression of DNA in cells across many organ systems [168,169,170,171].

#### 3.3.6. Magnetofection

Magnetofection is a nanoparticle-based gene delivery method that uses SPIONs to transport therapeutic genes to cells, tissues, and tumors. This approach has shown great potential in enhancing the efficacy of gene delivery up to several hundred-fold and in reducing the duration of gene delivery to minutes. To achieve this, SPIONs are complexed with DNA and exposed to an external magnetic field that guides them towards target cells.

Magnetofection is effective in vitro and in vivo, and it has been successfully used to achieve local transfection in the gastrointestinal tract and blood vessels. However, in vivo localization of SPIONs can be challenging, and the size of the particles can affect their ability to enter cells. Moreover, there are concerns over the cytotoxicity associated with the use of SPIONs [168,172,173].

#### 3.3.7. Naked RNA Injection

Naked mRNA offers an innovative approach to deliver genetic material without the need for traditional carrier molecules. This strategy entails the direct administration of mRNA through a process known as naked mRNA injection.

The allure of this approach lies in its simplicity, efficient preparation, cost-effectiveness, and facile storage. Notably, the technique has demonstrated its versatility through intramuscular injection in murine models, yielding diverse protein expression profiles. Similarly, human studies employing intradermal injection have showcased the successful translation of exogenous mRNA, underscoring its potential for human applications. Beyond these achievements, the intratumoral delivery of saline-formulated mRNA encoding four cytokines has emerged as a potent tool to effectively impede tumor growth.

However, despite these strides, the realm of naked mRNA is not devoid of challenges. Swift degradation within extracellular and intracellular environments by exo- and endonucleases poses a notable hurdle, significantly truncating the duration of protein translation stemming from the mRNA. Addressing this issue, innovative strategies such as packaging mRNA within the core of polyplex micelles and employing PEG-conjugated cationic polymers, have arisen. Remarkably, this approach confers substantial protection, shielding intact mRNA from degradation by over 10,000-fold compared to naked mRNA, even in conditions rich with ribonucleases [174,175,176,177,178].

#### 3.3.8. Naked DNA Injection

Naked DNA injection is the simplest and least expensive delivery method for gene therapy. This approach involves the direct injection of DNA at the targeted site. Although it offers the advantage of localized DNA uptake, the technique has several drawbacks. Poor and variable expression levels of the desired gene are often observed, and the method can cause damage to the tissue surrounding the injection site [168].

#### 3.3.9. Gene-Activated Matrix (GAM)

Gene therapy and tissue engineering can be combined to create a novel solution for repairing damaged tissues, known as the Gene-Activated Matrix (GAM). A GAM offers the potential to restore the structure and function of damaged or dysfunctional tissues. A GAM is a scaffold made of biomaterials that can be seeded with therapeutic genes to direct and sustain gene expression. It can be used for both in vivo and ex vivo approaches, providing a three-dimensional template for tissue regeneration.

While GAMs hold great potential, they may require other viral or non-viral vectors to increase expression, and DNA damage is possible during scaffold formation. Nonetheless, the ability to direct and sustain gene expression in a 3D template makes GAMs a promising candidate for tissue engineering and regenerative medicine [168,179].

#### 3.3.10. Hydrodynamic High-Pressure Injection

A hydrodynamic high-pressure injection is a gene delivery method that involves the rapid injection of a large volume of pDNA. This technique is simple, convenient, and highly efficient, making it a versatile tool for a variety of applications in gene therapy. Hydrodynamic injection uses a high-pressure flow of fluid to deliver DNA directly into the liver or other targeted tissues. The process is thought to work by briefly disrupting the plasma membrane, allowing the DNA to enter the cells.

One of the major advantages of hydrodynamic high-pressure injections is its simplicity. This method does not require specialized equipment or extensive training and can be easily performed by researchers with minimal experience in gene delivery. Additionally, hydrodynamic injections are highly efficient, with transfection rates as high as 60% in some studies. This makes it a useful tool for a variety of applications, including gene therapy, vaccine development, and gene function studies.

However, hydrodynamic injections do have some limitations. The high pressure used in this method can cause tissue damage, and the transient nature of the transgene expression limits its usefulness for long-term applications. Additionally, hydrodynamic injection is primarily useful for liver-specific gene expression, and its application to other tissues is still under investigation [180,181].

## 4. Twenty-Two Approved Human Gene Therapy Products

In this section, and in Table 3, our attention will be directed towards the culmination of gene therapy endeavors, namely the assortment of approved human gene therapy products that have successfully navigated the rigors of the market. These products have ushered gene therapy from a realm of promising potentiality to the realm of practical application. We shall endeavor to present an exhaustive inventory of these products, accompanied by an in-depth analysis of their underlying mechanisms, as well as an elucidation of their respective merits and limitations.

### 4.1. Approved Human Gene Therapy Products and Their Applications for In Vivo Treatment

#### 4.1.1. IMLYGIC/Melanoma, Pancreatic Cancer

IMLYGIC (Talimogenelaherparepvec) is an FDA-approved gene therapy product used to treat melanoma and pancreatic cancer in adults.

It utilizes the HSV-1 oncolytic virus vector, which has deletions in the y34.5 and a47 regions, and the GM-CSF gene is inserted into the deleted y34.5 loci. The therapy works by inducing tumor lysis and antitumor immune responses.

Priced at USD 65,000 per treatment, IMLYGIC has shown promise in clinical trials and has been approved by the USA FDA for use in patients [7,182,183].

#### 4.1.2. LUXTURNA/Retinal Dystrophy

Retinal dystrophy, caused by biallelic RPE65 mutation, leads to progressive vision loss and eventually blindness. However, LUXTURNA, a gene therapy product, has been approved by the USA FDA and EU FDA for the treatment of this condition.

LUXTURNA utilizes a recombinant adeno-associated virus (AAV2) vector to deliver a normal copy of the RPE65 gene to the retinal pigment epithelium (RPE) cells. This compensates for the biallelic mutation, allowing the RPE65 protein to convert trans-retinyl esters to 11-cis-retinal, the natural ligand and chromophore of the photoreceptors in the eye.

LUXTURNA is administered through a subretinal injection following a vitrectomy, and its application has shown significant improvement in visual function. The therapy is priced at USD 850,000 for both eyes or USD 425,000 per eye [7,184,185].

#### 4.1.3. Zolgensma/Spinal Muscular Atrophy

Spinal muscular atrophy (SMA) is a rare genetic disorder caused by mutations in the SMN1 gene, leading to the progressive degeneration of motor neurons and ultimately to muscle weakness and atrophy. Among SMA types, type I is the most severe and is often fatal. However, the approval of Zolgensma (Onasemnogene Abeparvovec) by the USA FDA offers new hope to pediatric patients less than 2 years of age with SMA type I.

Zolgensma is a gene therapy product that utilizes a non-replicating recombinant adeno-associated virus 9 (AAV9) vector to deliver a functional copy of the SMN1 gene under the control of the CMV enhancer/chicken-β-actin-hybrid promoter (CB) to express SMN1 in motor neurons of SMA patients. The AAV9 capsid is unique and has the ability to cross the blood–brain barrier, allowing efficient CNS delivery by intravenous administration. Furthermore, the AAV ITR modification in Zolgensma produces a self-complementary DNA molecule that forms a double-stranded transgene, enhancing active transcription.

Zolgensma offers a promising cure for SMA type I. However, it comes with a high price tag of USD 4.2–USD 6.6 million per patient. Nevertheless, the approval of Zolgensma is a significant milestone in the field of gene therapy [7,186].

#### 4.1.4. Spinraza/Spinal Muscular Atrophy

Another approved treatment available for spinal muscular atrophy (SMA) is Spinraza (Nusinersen), an ASO approved by the USA FDA and EU FDA. Spinraza targets intron 7 on the SMN2 hnRNA, modulating alternative splicing to increase the inclusion of exon 7 in the final processed RNA, resulting in higher levels of functional SMN protein in the central nervous system (CNS). Spinraza is indicated for both pediatric and adult patients with SMA. The price of Spinraza is USD 125,000 per injection [7,187].

#### 4.1.5. Patisiran/Polyneuropathy

Polyneuropathy, also known as familial amyloidotic polyneuropathy, is a progressive and fatal disease caused by the accumulation of abnormal protein in the nerves and organs. Hereditary transthyretin-mediated amyloidosis, also called ATTRv amyloidosis, is one type of polyneuropathy. Patisiran (Onpattro) is a novel treatment for ATTRv amyloidosis that was approved by the FDA in 2018 and the EU FDA in 2019.

Patisiran is an LNP containing an RNAi that targets the transthyretin (TTR) gene. When Patisiran is administered, it enters the cell and cleaves the transthyretin mRNA, leading to a reduction in the circulating transthyretin protein. This decrease in protein levels reduces the amyloid accumulation associated with transthyretin-mediated amyloidosis, ultimately slowing the progression of the disease.

Patisiran is administered through intravenous infusion at a recommended dose of 0.3 mg/kg once every three weeks. In clinical trials, Patisiran demonstrated a significant reduction in the progression of neuropathy and an improvement in quality of life. As of 2023, Patisiran is the only approved RNAi-based therapy for the treatment of a genetic disease. Patisiran is priced at USD 345,000 per 2 mg/mL [7,188].

#### 4.1.6. Gendicine/Carcinoma

Gendicine (rAd-p53) has been approved by the Chinese State Food and Drug Administration. Gendicine is a recombinant adenovirus vector that contains the wild-type p53 gene. The target of this therapy is the Tp53 tumor-suppressor gene, which is often mutated or lost in cancer cells. The adenovirus vector delivers the wild-type p53 gene into the cancer cells, where it inhibits cell proliferation and induces apoptosis. The mechanism of action of Gendicine involves the restoration of p53’s function in cancer cells.

Gendicine has been shown to be effective in preclinical and clinical trials, with a good safety profile. Its use in combination with other therapies has also been investigated. In terms of cost, Gendicine is relatively affordable, with a price of USD 387 per dose. However, its approval is currently limited to China [7,189,190].

#### 4.1.7. Neovasculgen/Atherosclerotic Peripheral Arterial Disease

Atherosclerotic peripheral arterial disease (PAD), including critical limb ischemia (CLI), is a debilitating disease that can lead to amputation of the affected limb. Neovasculgen, a pDNA therapy approved by the Russian Ministry of Healthcare, aims to promote angiogenesis and improve the blood flow to the affected area.

This therapy targets the 165-amino-acid isoform of human vascular endothelial growth factor (VEGF165) using a recombinant DNA construct that contains the necessary genetic information to produce the protein. Upon administration, the plasmid enters the cells and VEGF165 is produced, stimulating angiogenesis, endothelial migration, and cellular proliferation. The DNA construct contains a transcription start site, splicing signal, and transcription terminator, as well as a polyadenylation signal to ensure proper mRNA processing. The therapy is administered via transmuscular transfer to the calf muscles. The price of the treatment course is USD 6600 [7,191].

#### 4.1.8. Oncorine/Nasopharyngeal Cancer

Nasopharyngeal cancer, a type of head and neck cancer, along with lung, liver, and pancreatic cancers, has been targeted by an oncolytic adenovirus type 5 product called Oncorine, which has been approved by the Chinese State Food and Drug Administration.

The vector of Oncorine is a replication-selective adenovirus with the E1B-55 kDa gene entirely deleted. This gene is responsible for the inactivation of the p53 tumor-suppressor gene. The mechanism of action is based on the selective replication of Oncorine in p53-deficient cancer cells, leading to their lysis. Oncorine propagates selectively in cancer cells, while normal cells do not get infected because of the lack of E1b-55KD in the adenovirus, thereby ensuring the safety of the product. Subsequently, Oncorine-mediated cell cytotoxicity is initiated, and the adenoviruses released from lysed cancer cells can infect neighboring cancer cells, triggering a cascade of cancer cell death [7,192,193].

#### 4.1.9. Defitelio/SOS/VOD

Defibrotide, also known as Defitelio or defibrotide sodium, is a gene therapy product that has been approved by the USA FDA and EU FDA for the treatment of a rare and life-threatening disease known as sinusoidal obstruction syndrome (SOS) or veno-occlusive disease (VOD) with multiorgan dysfunction. Defibrotide is a combination of single-stranded oligo DNAs with aptameric functions that are obtained from porcine mucosa tissue through controlled depolymerization.

The mechanism of action of Defibrotide involves its ability to act on vascular endothelial cells. It binds to the endothelial cell membrane and prevents thrombosis and inflammation, thereby promoting fibrinolysis and reducing endothelial cell damage. The aptameric function of Defibrotide allows it to act as an anticoagulant and also exhibit anti-inflammatory properties. Defibrotide costs USD 7425 per day [7,192,193,194].

#### 4.1.10. ADSTILADRIN/Cancer (BCG-NMIBC)

One of the most recent breakthroughs in this field has been the development of ADSTILADRIN (nadofaragene firadenovec-vncg), a gene therapy product approved by the USA FDA in December 2022. ADSTILADRIN is specifically designed for the treatment of bacillus Calmette–Guérin (BCG)-unresponsive non-muscle invasive bladder cancer (NMIBC) with carcinoma in situ (CIS) with or without papillary tumors in adults.

ADSTILADRIN is a non-replicating adenoviral vector-based gene therapy that utilizes a recombinant adenovirus serotype 5 vector. The vector is designed to deliver a copy of a gene encoding human interferon-alfa 2b (IFNα2b) to the bladder urothelium. Upon intravesical instillation, ADSTILADRIN results in cell transduction and transient local expression of the IFNα2b protein, which is anticipated to have anti-tumor effects.

One of the advantages of ADSTILADRIN is its targeted delivery system, which allows for precise treatment of the affected area while minimizing systemic side effects. The therapy’s mechanism of action is centered around the localized expression of IFNα2b, a protein known to have anti-tumor effects. While the treatment is only approved for BCG-unresponsive non-muscle invasive bladder cancer (NMIBC) with carcinoma in situ (CIS) with or without papillary tumors, it is expected that the technology underlying ADSTILADRIN may be applicable to a wider range of cancers [195,196].

#### 4.1.11. HEMGENIX/Hemophilia B

Hemophilia B is caused by a deficiency in clotting factor IX and is characterized by bleeding episodes that can lead to joint damage and other serious health issues. In November 2022, the US FDA approved HEMGENIX, an adeno-associated virus serotype 5 (AAV5)-based gene therapy developed by CSL Behring.

HEMGENIX delivers a copy of a gene encoding the Padua variant of human coagulation factor IX (hFIX598 Padua) to the liver cells. A single intravenous infusion of HEMGENIX leads to cell transduction and an increase in circulating factor IX activity in patients with hemophilia B. This therapy has shown significant potential in providing long-term benefits for adult patients with hemophilia B [197,198,199].

#### 4.1.12. Upstaza/AADC Deficiency

Aromatic L-amino acid decarboxylase (AADC) deficiency is a rare genetic disorder caused by mutations in the gene responsible for producing the AADC enzyme, which is essential for the production of dopamine. Dopamine is a neurotransmitter that plays a critical role in movement control, and patients with AADC deficiency experience very little or no dopamine production in the brain.

Fortunately, a new gene therapy product called Upstaza (eladocagene exuparvovec) has been approved by the EU FDA as of July 2022 for the treatment of AADC deficiency in adults and children aged 18 months and older.

The product uses adeno-associated virus (AAV) as a vector to carry the functional AADC gene into nerve cells. This enables the cells to produce the missing enzyme, which in turn leads to the production of the dopamine necessary for proper brain function and an improvement in the symptoms of the condition [200,201].

### 4.2. Approved Human Gene Therapy Products and Their Applications for Ex Vivo Treatment

#### 4.2.1. ABECMA/Multiple Myeloma

Multiple myeloma is a type of blood cancer that affects plasma cells. It occurs when these cells become abnormal and grow uncontrollably, leading to the production of excessive amounts of abnormal protein. The USA FDA and EU FDA have approved ABECMA (idecabtagene vicleucel), a gene therapy for multiple myeloma.

ABECMA is an autologous human T cell gene therapy that uses a lentiviral vector (LVV) to introduce a chimeric antigen receptor (CAR) that recognizes B cell maturation antigen (BCMA) in the patient’s own T cells. Once modified, these T cells are infused back into the patient, where they recognize and kill the cancerous plasma cells. ABECMA is approved for use in adults who have not responded to other types of treatment or who have relapsed after previous treatments [202,203,204,205].

#### 4.2.2. CARVYKTI/Multiple Myeloma

CARVYKTI (ciltacabtagene autoleucel) is a gene therapy product approved by both the USA FDA and EU FDA for the treatment of adults with multiple myeloma. The product uses a lentiviral vector (LVV) to introduce an anti-B cell maturation antigen (BCMA) chimeric antigen receptor (CAR) into the patient’s own T cells. After being genetically modified ex vivo, the T cells can target and destroy the cancer cells expressing BCMA.

The therapy involves extracting the patient’s T cells and genetically modifying them with the lentiviral vector encoding the anti-BCMA CAR, which is then infused back into the patient. The CAR T cells recognize the cancer cells expressing BCMA and destroy them [206,207,208,209].

#### 4.2.3. SKYSONA/Cerebral Adrenoleukodystrophy (CALD)

Cerebral adrenoleukodystrophy (CALD) is a rare neurodegenerative disease that affects boys aged 4–17. SKYSONA (elivaldogene autotemcel) is an autologous hematopoietic stem cell (HSC) gene therapy that was approved by the USA FDA in September 2022 for the treatment of CALD through an accelerated approval pathway.

SKYSONA is prepared from the patient’s own HSCs, which are collected via apheresis procedures. The cells are then transduced ex vivo with Lenti-D LVV, which is a replication-incompetent, self-inactivating lentiviral vector (LVV) that carries a functional ABCD1 gene that encodes normal adrenoleukodystrophy protein (ALDP). The ABCD1 gene is under the control of an internal MNDU3 promoter, which is a modified viral promoter that has been shown to effectively control the expression of the transgene in HSCs and their progeny in all lineages.

SKYSONA has been shown to slow the progression of neurologic dysfunction in patients with early, active CALD. The therapy works by replacing the faulty gene with a functional one, which can produce the missing protein that is needed for proper brain function. SKYSONA is administered via a single infusion and has shown promising results in clinical trials, with patients demonstrating significant improvements in neurological function [210,211,212].

#### 4.2.4. YESCARTA/Large B Cell Lymphoma

The advent of cancer immunotherapy has transformed the treatment landscape for certain types of cancer. One such example is YESCARTA (axicabtagene ciloleucel), a gene therapy product that was approved by the FDA in October 2017 for the treatment of adults with certain types of cancer, specifically, relapsed or refractory large B cell lymphoma after two or more lines of systemic therapy, including diffuse large B cell lymphoma (DLBCL) not otherwise specified, primary mediastinal large B cell lymphoma, high-grade B cell lymphoma, and DLBCL arising from follicular lymphoma. YESCARTA is an autologous CAR T cell therapy that uses a gamma-retroviral vector to transfect anti-CD19 CAR genes into the patient’s T cells.

The ex vivo modulated autologous T cells are transfected with the gamma-retroviral vector, which expresses a chimeric antigen receptor (CAR) consisting of a murine anti-CD19 single-chain variable fragment and co-stimulatory domains of CD28 and CD3-zeta. After collecting autologous T cells from patients through leukapheresis, the cells are then enriched in a closed system in the YESCARTA manufacturing center. The transfected T cells are activated with anti-CD3 and IL-2, transduced with a retroviral vector expressing the anti-CD19 CAR gene, and then infused back into the patient after less than 10 days of manufacturing processes. The CAR T cell infusion targets and eliminates CD19-expressing cancer cells in the patient.

YESCARTA is priced at USD 373,000 per dose. While YESCARTA was initially approved for the treatment of large B cell lymphoma, ongoing clinical trials are also investigating the potential of YESCARTA in the treatment of other types of cancer, such as melanoma and pancreatic cancer [7,213,214].

#### 4.2.5. KYMRIAH/Lymphoma

KYMRIAH (tisagenlecleucel CTL019) is a genetically engineered product composed of autologous T cells that have been genetically modified with a lentiviral vector to produce a CAR consisting of a murine single-chain antibody fragment (scFv) that is specific for CD19, which is linked to an intracellular cytoplasmic domain for 4-1BB (CD137) and CD3 zeta with a CD8 transmembrane hinge. The FDA and the EU have approved KYMRIAH for the treatment of pediatric and young adult patients up to 25 years of age with relapsed or refractory follicular lymphoma.

The lentiviral vector used in the manufacturing process of KYMRIAH is pseudotyped with a VSV g envelope derived from the HIV-1 genome in ex vivo conditions. This vector integrates into the genome of the transduced cells, resulting in the production of the tisagenlecleucel CAR under the regulation of a constitutively active promoter. When the tisagenlecleucel CAR binds to target cells, which are CD19-expressing cells in this case, it initiates antitumor activity through the CD3 domain. The intracellular 4-1BB costimulatory domain enhances the antitumor response and also ensures durable persistence of the CAR T cells. The price of KYMRIAH is approximately USD 475,000 per dose [6,7,166,215,216,217].

#### 4.2.6. Strimvelis/ADA-SCID

Strimvelis (GSK-2696273) is a gene therapy product approved by the EU FDA in May 2016 to treat patients with severe combined immunodeficiency due to adenosine deaminase deficiency (ADA-SCID), a rare inherited condition in which patients lack the ADA enzyme required to maintain healthy lymphocytes, leading to a dysfunctional immune system. This disease is life-threatening, and without effective treatment, patients rarely survive more than two years. Strimvelis is used for patients who cannot undergo a bone marrow transplant due to the unavailability of a suitable, matched, related donor.

The product contains cells derived from the patient’s own bone marrow, including CD34+ cells genetically modified to contain a working ADA cDNA gene. This retroviral-based vector therapy functions by delivering genes into the patient’s body to produce the necessary protein. The medicine has been approved for adult use, and the cost of treatment is EUR 163,900 [7,218].

#### 4.2.7. BREYANZI/Large B Cell Lymphoma (LBCL)

Large B cell lymphoma (LBCL) is a type of blood cancer that affects B cells, a type of white blood cell that produce antibodies. B cell lymphomas are the most common type of non-Hodgkin’s lymphoma (NHL) and can be aggressive or indolent. BREYANZI (lisocabtagene maraleucel) is a type of gene therapy that is used to treat adult patients with LBCL who have not responded to or have relapsed after at least two other types of therapy. It is a type of CAR T cell therapy that uses the patient’s own T cells to fight the cancer.

BREYANZI works by using a lentiviral vector to genetically modify the patient’s T cells to express a chimeric antigen receptor (CAR) that targets CD19, a protein found on the surface of B cells. The CAR is made up of CD8+ and CD4+ cell components, which work together to recognize and attack the cancer cells. The modified T cells are then infused back into the patient, where they multiply and attack the cancer cells.

BREYANZI was approved by the US FDA in 2021 and the EU FDA in 2022. It is the first CAR T cell therapy that can be given as a single infusion [219,220,221].

#### 4.2.8. TECARTUS/Acute Lymphoblastic Leukemia (ALL)

Acute lymphoblastic leukemia (ALL) is a devastating disease that affects adults. Fortunately, TECARTUS (brexucabtagene autoleucel), a gene therapy product, has been approved by both the USA FDA and the EU FDA to combat this disease.

TECARTUS uses a retroviral vector to genetically modify autologous T cells ex vivo with an anti-CD19 chimeric antigen receptor (CAR) consisting of a murine anti-CD19 single-chain variable fragment (scFv) linked to a CD28 co-stimulatory domain and a CD3-zeta signaling domain. The mechanism of action of TECARTUS is the targeting of CD19-positive cells in the patient, which is achieved through the expression of the anti-CD19 CAR on the modified T cells. The modified T cells are then infused back into the patient where they multiply and attack the CD19-positive cells. The price of TECARTUS is USD 373,000 [217,222,223,224].

#### 4.2.9. ZYNTEGLO/Beta-Thalassemia

Beta-thalassemia is a genetic disorder that affects the production of hemoglobin, causing anemia and other complications. ZYNTEGLO (betibeglogene autotemcel) is a promising gene therapy product for this disease, which has been approved by the USA FDA and EU FDA. This product uses a Lentiglobin BB305 lentiviral vector to deliver the beta-A-T87Q-globin gene to autologous CD34+ cells enriched with hematopoietic stem cells. This process enhances the production of hemoglobin and improves the quality of life for patients with beta-thalassemia. It is a one-time treatment and has demonstrated significant efficacy in clinical trials.

The approval of ZYNTEGLO marks a significant milestone in the field of gene therapy, as it is the first approved gene therapy product for beta-thalassemia. This product provides hope to patients suffering from this debilitating disease, offering a potentially curative treatment option. However, the high cost of ZYNTEGLO, currently priced at USD 1.8 million, has sparked discussions regarding the affordability and accessibility of gene therapies. Despite this, the approval of ZYNTEGLO represents a major step forward for gene therapy and offers renewed optimism for the development of future gene therapies for genetic diseases [225].

#### 4.2.10. Libmeldy/Metachromatic Leukodystrophy

Leukodystrophy is a rare and debilitating disease that affects the nervous system in children. A new gene therapy called Libmeldy has been developed and was approved by the EU FDA in December 2020 to help treat the condition. Libmeldy is an autologous ex vivo lentiviral gene therapy that uses a type of virus called a lentivirus to introduce a healthy copy of the ARSA gene into CD34+ cells extracted from the patient’s blood or bone marrow. These CD34+ cells are then infused back into the patient’s bloodstream where they migrate to the bone marrow and begin producing functional ARSA.

The ARSA protein helps to break down sulfatides, which build up in the nervous system of patients with metachromatic leukodystrophy. By introducing healthy ARSA-producing cells into the patient, Libmeldy helps to control the symptoms of the disease, with long-lasting effects expected [226].

## 5. Future Directions

As we cast our gaze towards the horizon, the landscape of gene therapy reveals a realm brimming with potentialities yet to be harnessed. With a deepening comprehension of the intricacies of the human genome and the maturation of our genetic manipulation tools, the vista of gene therapy seems poised for continued expansion. The path ahead is one of unwavering research and innovative strides, a journey that holds the promise of ushering forth more potent treatments and transformative remedies for a diverse spectrum of ailments. The transformative impact on the lives of patients globally is on the precipice of realization. For instance, a paramount obstacle on the path to clinical translation resides in the nonspecific sequestration of gene-therapeutic-loaded delivery carriers by reticuloendothelial system (RES) organs, notably liver sinusoidal scavenger wall cells. This not only compromises targeted delivery but also accentuates concerns of toxicity and immunogenicity. Stealth coating with PEG-oligopeptides has emerged as a strategic intervention, transiently and selectively obstructing the scavenging functions of liver sinusoidal wall cells. This breakthrough approach circumvents sinusoidal capture, thus elevating gene transfer efficacy within target tissues. Moreover, the lingering challenge of undesired hepatic off-target accumulation of FDA-approved mRNA-encapsulated LNPs persists, even following local intramuscular administration. This inadvertently triggers adverse hepatic side effects and inflammation, constraining their broad application. A paradigm shift beckons, characterized by minimizing hepatic off-targeting while optimizing injection site delivery. Such precision targeting not only augments vaccine applications but also holds the promise of aiding patients beset by systemic inflammation. The confluence of innovative solutions with the pursuit of precision heralds the dawn of an era where gene therapy’s potential can reach unprecedented heights [226,227,228,229,230,231].

## 6. Conclusions

Our in-depth analysis of the current state of human gene therapy elucidates the significant progress and the transformative potential of this field. The range of approved in vivo and ex vivo products, such as IMLYGIC, LUXTURNA, Zolgensma, Spinraza, Patisiran, Gendicine, Neovasculgen, Oncorine, Defibrotide, ADSTILADRIN, HEMGENIX, Upstaza, ABECMA, CARVYKTI, SKYSONA, YESCARTA, KYMRIAH, Strimvelis, BREYANZI, TECARTUS, ZYNTEGLO, and Libmeldy, demonstrates the growing applicability of gene therapy across various diseases, including melanoma, pancreatic cancer, retinal dystrophy, spinal muscular atrophy, polyneuropathy, hereditary transthyretin-mediated amyloidosis, head and neck squamous cell carcinoma, atherosclerotic peripheral arterial disease, critical limb ischemia, nasopharyngeal cancer, SOS/VOD with multiorgan dysfunction, bacillus Calmette–Guérin (BCG)-unresponsive non-muscle invasive bladder cancer, hemophilia B, aromatic L-amino acid decarboxylase deficiency, multiple myeloma, cerebral adrenoleukodystrophy, lymphoma, ADA-SCID, large B cell lymphoma, acute lymphoblastic leukemia, beta-thalassemia, and metachromatic leukodystrophy.

Gene therapy has emerged as a robust and versatile tool poised to catalyze the advancement of personalized medicine, revolutionizing therapy efficacy and ameliorating the collateral effects of widely used pharmaceutical agents. The most rapid evolution of personalized medicine is witnessed in the realms of cancer treatment and hereditary disorders, where gene therapy strategies and products play a pivotal role. However, the potential of gene therapy extends far beyond these domains, encompassing a diverse array of pathologies, such as neurodegenerative ailments, immune disorders, inflammatory processes, and more, calling for further investigation.

Despite remarkable progress, challenges remain in the practical use of gene therapy. These include, but are not limited to, issues related to safety, specificity, delivery efficiency, and immune responses. For instance, while viral vectors are highly efficient, they can provoke immune responses. Non-viral vectors, while less immunogenic, are less efficient. Balancing these contrasting attributes is a key concern.

Moreover, certain ethical, regulatory, and commercialization challenges must be addressed for gene therapy to become more widely adopted. Researchers embarking on the nascent phases of product development are tasked with a multifaceted problem encompassing current technological frontiers, product commercialization, and the potential for costly solutions. While some of these innovative solutions hold transformative promise, their accessibility for patients could be constrained by elevated costs. It is imperative that these pioneering commercial decisions are underpinned by judicious forethought, as ill-conceived approaches could jeopardize not only production but also the integration of gene therapy products into medical practice. Given the potentially transformative impact of gene therapy, it is critical that these challenges are met with thoughtful and comprehensive solutions.

Looking forward, the field of gene therapy holds immense potential. As our understanding of the human genome deepens and our tools for genetic manipulation become more sophisticated, the scope of gene therapy will likely continue to broaden. Continued research and innovation in this field will undoubtedly lead to more effective treatments and cures for a range of diseases, transforming the lives of patients worldwide.

It is our hope that this review provides researchers, clinicians, and policymakers with a comprehensive understanding of the current state of human gene therapy, serving as a launchpad for future advancements in this promising field.

## Figures and Tables

**Figure 1 pharmaceuticals-16-01416-f001:**
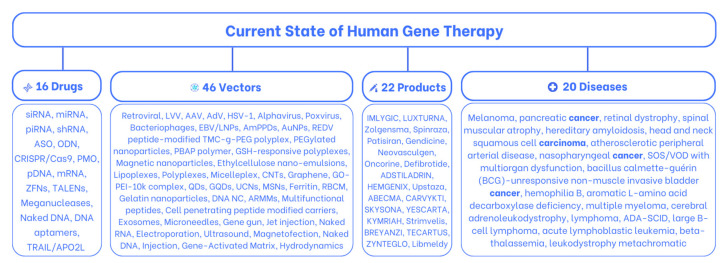
The current state of human gene therapy: 16 drugs, 46 carriers, and 22 approved products that treat 20 diseases.

**Table 1 pharmaceuticals-16-01416-t001:** Gene therapy drugs.

Drug	Full Name	Short Description, Functions	Size	Pros	Cons
siRNA	Short interfering RNA or silencing RNA	siRNA with a complementary sequence can be utilized to silence any gene. The functions of siRNA include RNA cleavage, regulation of protein-coding genes, and transposons. Additionally, siRNA plays a role in antiviral defense.	20–25 bp	Exceptional target specificity due to its perfect complementarity. Promising results in delivering drugs to the brain.	Off-target effects and activation of the innate immune response [9,10,11,12,13,14,15].
miRNA	microRNA	miRNAs are small non-coding RNAs that regulate gene expression post-transcriptionally through translational repression, mRNA degradation, and protein-coding gene regulation.	~22 bp	Plant miRNA and target mRNA often show a high degree of complementarity. In animals, miRNAs have diverse functions, including developmental timing, tissue growth, and apoptosis.	Animal miRNA–target duplexes have more structural variability with gaps and mismatches in the short complementary sequence stretches. However, specific rules for functional miRNA–target pairing are yet to be established despite known functional targets [11,16].
piRNA	PIWI-interacting RNA	piRNA performs diverse functions such as transcriptional or post-transcriptional repression of transposons, multigenerational epigenetic phenomena in worms, and pre-pachytene piRNA-mediated transposon silencing by binding to PIWI proteins.	26–32 bp	Diverse functions: transcriptional or post-transcriptional repression of transposons, multigenerational epigenetic phenomena in worms, and pre-pachytene piRNA-mediated transposon silencing by binding to PIWI proteins.	Further research and clinical investigations are warranted [15,17].
shRNA/Hairpin Vector	A short hairpin RNA or small hairpin RNA	The Hairpin Vector, or shRNA, is an engineered RNA molecule with a sharp turn that can silence gene expression through RNA interference.	19–29 bp	Precise and enduring gene silencing, along with a slow rate of degradation and turnover.	In medicinal applications, the use of an expression vector may cause side effects [18,19,20,21,22].
ASO	Antisense oligonucleotide	ASOs are short DNA or RNA strands that can prevent protein synthesis by binding to complementary mRNA strands.	18–30 bp	Effective targeting of lncRNAs in both nuclear and cytoplasmic locations.	Potential off-target effects and suboptimal efficacy [23,24,25,26,27,28,29].
ODNs	Oligodeoxynucleotides	ODNs can silence genes using the antisense or antigene strategy.	can vary	Easy synthesis and manipulation and the ability to target specific tissue transcription factors.	A short lifetime due to high degradation rates from endocytosis or nucleases [30].
CRISPR/Cas9	Clustered regularly interspersed short palindromic repeats (CRISPR)/CRISPR-associated protein 9 (Cas9), formerly called Cas5, Csn1, or Csx12	CRISPR/Cas9 is a bacterial protein that cuts DNA, altering a cell’s genome and serving as a defense mechanism against DNA viruses and plasmids.	can vary	Simple design, delivery through plasmids or viral vectors. Off-target editing occurs frequently without additional homologous sequence.	Multiplexing difficulties and dependence on PAM for targeting [31,32,33,34,35].
pDNA	Plasmid DNA	An extrachromosomal DNA molecule in cells that can replicate independently and is physically distinct from chromosomal DNA.	up to >106 Da	No substantial formulation or modification of plasmid molecules is necessary. Has potential use in treating cardiovascular disease, as transfer to cardiac or skeletal muscle is possible.	Immunogenicity. Plasmid molecules necessitate nuclear entry subsequent to cytoplasmic ingress [36,37,38,39,40,41,42,43,44,45].
mRNA	A messenger ribonucleic acid	mRNA delivers correct genetic information to cells for healthy protein synthesis, correcting gene mutations that result in damaged proteins and diseases.	can vary	Simple to modify, quick and temporary protein production, and adaptable without causing mutations.	Its potential to trigger immune responses and the difficulty in regulating the concentration of reporter mRNA [46,47,48].
Meganucleases	Meganucleases	Meganucleases are highly specific endodeoxyribonucleases that act as molecular DNA scissors, characterized by a large recognition site. They can be used to replace, eliminate, or modify sequences in a highly targeted way, and are considered the most specific naturally occurring restriction enzymes.	18–30 bp	Tolerance of some mismatches, low off-target editing, and a small size that enables their use with a variety of viral vectors.	Challenging reengineering for novel specificities and design complexity [49,50,51,52,53].
ZFNs	Zinc Finger Nucleases	ZFNs are artificial endonucleases combining a zinc finger protein (ZFP) with the cleavage domain of the FokI restriction enzyme.	can vary	Tolerating some mismatches and showing low to moderate off-target editing.	Difficulty with G-rich regions, challenging design, and limited multiplexing [54,55].
TALENs	Transcription activator-like effector nucleases	TALENs are custom restriction enzymes designed for precise DNA cleavage.	32–40 bp	Mismatches allowed, low off-target editing, moderate design complexity.	The need for a 5’ T for each TALEN, difficult multiplexing, and large size that limits viral vectors due to repetitive sequences leading to unwanted recombination [56,57,58,59,60].
DNA aptamers	DNA aptamers	DNA aptamers are artificial DNA sequences with high affinity to specific target molecules due to their structural conformations.	10–50 bp	Ease of generation and synthesis, stability, low immunogenicity, efficient penetration, less batch variation, easy modification, cost-effectiveness, and short production times.	Further research and clinical investigations are warranted [61,62,63,64,65,66,67,68,69,70].
TRAIL/APO2L	Tumor necrosis factor-related apoptosis-inducing ligand	TRAIL/APO2L is a member of the tumor necrosis factor family that induces apoptosis of tumor cells through binding to death receptor 4 or 5 without affecting normal tissues.	can vary	Highly selective for inducing apoptosis in tumor cells while sparing normal cells and tissues.	The presence of TRAIL-resistant cancer cells [71,72].
PMO	Phosphorodiamidate Morpholino Oligomer	PMO are single-stranded DNA analogs that block protein translation by binding to complementary sequences of target mRNA through steric blockade. They consist of morpholine rings connected by phosphorodiamidate linkages.	6–22 bp	Resistant to biologic enzymes, ideal for in vivo applications.	Further research and clinical investigations are warranted [73,74].
Naked DNA	Naked DNA	Direct injection of naked DNA can transfer genes to various tissues, including skin, thymus, cardiac muscle, skeletal muscle, and liver.	2–19 kb	Naked DNA has a simple and safe method suitable for certain applications like DNA vaccination, with long-term expression observed in skeletal muscle for over 19 months.	Low gene delivery efficiency, with a single injection resulting in transgenic expression in less than 1% of muscle fibers. However, multiple injections can improve efficiency [75].

**Table 2 pharmaceuticals-16-01416-t002:** Gene therapy carriers.

Vector	Type	Short Description	Pros	Cons
Retroviral	Viral vector	RNA viruses that use a DNA intermediate for replication.	Suitability for ex vivo gene therapy, such as for transducing CD34+ bone marrow hematopoietic stem cells or peripheral blood lymphocytes.	Low vector concentration, in vitro transfection inefficiency, particle instability, inability to transduce non-dividing post-mitotic cells, and particles that only infect proliferating cells [75,79,80].
Lentiviral vector (LVV)	Viral vector	A single-stranded RNA-based spherical structure.	Long-lasting gene transfer in proliferating cells.	The potential for non-target mutations and the risk of oncogenesis due to integration in both dividing and nondividing cells [75,79,80].
Adeno-associated virus (AAV)	Viral vector	AAV, a non-enveloped DNA virus, boasts favorable biological features in gene therapy.	AAV is a stable, efficient, and easily scalable double-stranded DNA virus. It has high gene transduction rates and is non-immunogenic and non-pathogenic.	Expensive, complex to produce, limited packaging capacity [75,80,81].
Adenovirus (AdV)	Viral vector	A non-enveloped virus with a double-stranded DNA and an icosahedral nucleocapsid.	Simple purification, genetic stability, large loading capacity for foreign genes, and effective transduction of various tissues.	Potential for strong inflammatory response [7].
Herpesviral vector (HSV-1)	Viral vector	HSV-1, a human virus-causing labial herpes, has unique properties used for vector design, particularly in CNS disease therapy.	High loading capacity; neuron-specific targeting.	Potential for strong inflammation; transient gene expression outside neurons [80,82].
Alphavirus	Viral vector	Alphaviruses are enveloped single-stranded RNA viruses, consisting of a protein capsid structure and spike membrane proteins. They recognize laminin and heparin receptors on mammalian and insect cells to deliver the RNA genome to the cell cytoplasm for immediate replication.	Wide host range and transient high expression of foreign genes.	Costly virus stock production; temporary gene expression [83,84].
Poxvirus	Viral vector	A family of double-stranded DNA viruses, with 83 species among 22 genera, divided into two subfamilies. Smallpox is a disease associated with this family.	Prolonged immunological effects, and promising results as oncolytic therapy in early clinical trials for various cancers.	Undeveloped therapeutic potential for most species [85,86,87,88].
Bacteriophages	Viral vector	Bacteriophages, also known as phages, are viruses that specifically target and replicate within bacterial cells. They are present everywhere and are considered the most prevalent biological agent on the planet. Phages consist of a genetic material surrounded by a shell of capsid proteins.	Advantages of bacteriophages include their specificity, low cost of production, stability, ability to package non-phage materials, ease of modification, and safety as generally recognized as safe (GRAS) agents.	Preparation of clinical formulations and mitigation of the risk of bacterial resistance via genetic material transmission remain unresolved. Additional focused investigations are imperative to address these concerns prior to the safe utilization of BPs in human subjects [89,90,91,92,93,94,95].
Epstein–Barr virus (EBV)/Human gammaherpesvirus 4	Viral vector	EBV, also known as human gammaherpesvirus 4, is a DNA virus.	EBV vector allows for controlled and long-term exogenous gene expression in neurons without integration into the host genome.	Further research and clinical investigations are warranted [96,97,98,99,100].
Lipid nanoparticles (LNPs)	Non-viral	Lipid nanoparticles (LNPs) encapsulate both fat-soluble and water-soluble drugs and deliver genetic drugs to the body through fusion with cell membranes. They are formed by the orientation of phospholipid biomolecules.	Biocompatible, reduce drug toxicity, overcome drug resistance, and promote endosomal release.	Further research and clinical investigations are warranted [75,101].
Amphiphilic phospholipid peptide dendrimers (AmPPDs)	Non-viral	Amphiphilic phospholipid peptide dendrimers (AmPPDs) compact siRNA into nanoparticles using natural lipid derivatives to protect against enzymatic degradation.	AmPPD advantages include an effective siRNA delivery system for anticancer therapy, and their simplicity in synthesis and versatility of functionalization.	Limited siRNA loading capacity and premature payload leakage [102].
Gold nanoparticles (AuNPs)	Non-viral	Gold nanoparticles (AuNPs) are inorganic biomaterials suitable for drug delivery and molecular diagnostics due to their easy synthesis and surface modification capabilities. They can efficiently attach various biomacromolecules and are used as drug carriers and molecular nanoprobes for detection and monitoring of target molecules.	Gold nanoparticles (AuNPs) are non-toxic and non-immunogenic, with efficient biomolecule attachment through various methods. They have versatile applications as drug carriers and molecular nanoprobes due to their unique optical properties and facile surface modification.	Further research and clinical investigations are warranted [103,104,105,106,107,108].
REDV peptide-modified TMC-g-PEG polyplex	Non-viral	A REDV peptide-modified TMC-g-PEG polyplex was developed for targeted delivery of miRNA-126 to VECs. The polyplex comprised of TMC, PEG and REDV peptide linked to TMC via PEG.	Could potentially be used as a miRNA carrier in artificial blood vessels for rapid endothelialization.	Further research and clinical investigations are warranted [109].
PEGylated nanoparticles	Non-viral	PEGylation refers to the coating of nanoparticles with polyethylene glycol (PEG).	Advantages of PEGylated nanoparticles include an enhanced systemic circulation time and reduced immunogenicity. The PEG coating protects the surface from aggregation, opsonization, and phagocytosis, leading to a prolonged circulation time in the body.	Further research and clinical investigations are warranted [110].
PBAP polymer (polybutylene adipate terephthalate polymer)	Non-viral	PBAP polymer, a biodegradable random copolymer, is made from adipic acid, 1,4-butanediol, and terephthalic acid.	Completely degradable. Efficiently delivers DNA, mRNA, Cas9 RNP, and S1mplex with low toxicity and high transfection efficacy.	Further research and clinical investigations are warranted [111,112,113,114,115,116].
GSH-responsive polyplexes	Non-viral	GSH-responsive polyplexes form nanoplatforms for effective delivery of negatively charged genetic biomacromolecules, such as DNA, mRNA, and Cas9/sgRNA RNP, improving cellular uptake and endosomal escape.	Efficient delivery of genetic biomacromolecules, low cytotoxicity, and ease of surface functionalization.	Further research and clinical investigations are warranted [117].
Magnetic nanoparticles	Non-viral	Magnetic nanoparticles are a class of nanoparticles that can be manipulated using magnetic fields. Such particles commonly consist of two components: a magnetic material, often iron, nickel or cobalt, and a chemical component that has functionality.	Intrinsically biocompatible. Magnetic moments can be manipulated using externally applied magnetic fields to control and leverage their nanoscale behavior. Ease of synthesis and modification.	Further research and clinical investigations are warranted [118,119,120,121,122,123,124].
Ethylcellulose nano-emulsions	Non-viral	Ethylcellulose nano-emulsions are utilized to create folate–ethylcellulose nanoparticle complexes through a low-energy method with aqueous components.	Biocompatible and plentiful, classified by FDA as generally recognized as safe (GRAS), and can have a positive or negative surface charge.	Further research and clinical investigations are warranted [125].
Lipoplexes	Non-viral	Lipoplexes consist of cationic lipids that have a hydrophilic and hydrophobic region. The lipids have a charged headgroup and a hydrocarbon chain or cholesterol derivative attached by a linker, such as glycerol.	Simple manufacturing process and do not typically cause immune responses like viral vectors.	Further research and clinical investigations are warranted [126,127].
Polyplexes	Non-viral	Polyplexes are formed by electrostatic interactions between cationic polymers and genes, resulting in nanoscale complexes. These complexes contain condensed and/or complexed gene or siRNA.	Polyplexes have advantages such as low toxicity, low immunogenicity, high versatility, and biodegradability. They can form complexes with small RNAs, protecting them and facilitating intracellular release.	Further research and clinical investigations are warranted [128,129].
Micelleplex	Non-viral	Micelleplexes, nanostructured NA/micelle-like complexes, draw their unique identity from the distinct chemical composition of amphiphilic copolymers, featuring domains that are cationic and hydrophilic/hydrophobic, thus facilitating the condensation of nucleic acids with one or more cationic blocks.	Effective binding, transportation, and targeted delivery of nucleic acids to cancer cells. The micelleplexes outperform their polyplex counterparts regarding gene silencing, internalization, toxicity, colloidal stability, and payload trafficking.	Nanosystems characterized by excessive positive charge density elicit substantive concerns pertaining to in vivo toxicity, necessitating critical appraisal [130,131]
Carbon nanotubes (CNTs)	Non-viral	CNTs are cylindrical carbon allotropes produced mainly via chemical vapor deposition.	CNTs have high surface areas, strength, biocompatibility, and can release therapeutic agents at specific locations. They also have a needle-like structure, flexible interaction with cargo, and a high drug loading capacity. Additionally, they possess outstanding optical and electrical properties and are highly stable.	CNTs have potential toxicity concerns in biological applications, necessitating careful evaluation of their safety profile [132].
Graphene	Non-viral	Graphene is a single layer of carbon atoms arranged in a hexagonal honeycomb lattice. It has a molecular bond length of 0.142 nanometers and is stacked to form graphite. Van der Waals forces hold separate graphene layers together, which can be overcome during exfoliation.	Thinnest and lightest known material with exceptional strength.	Potential toxicity and improper release of therapeutic agents [133,134].
GO-PEI-10k complex	Non-viral	The GO-PEI-10k complex combines graphene oxide with polyethyleneimine (PEI) to form a stable, positively charged complex that can bind with pDNA for intracellular transfection of the enhanced green fluorescence protein (EGFP) gene in HeLa cells, with reduced toxicity compared to bare PEI-10k polymer.	Minimal toxicity to cells and effective in gene transfer.	Further research and clinical investigations are warranted [134].
Quantum dots (QDs)	Non-viral	Quantum dots (QDs) are nanocrystals with unique optical and electronic properties arising from quantum mechanics, which make them useful for various biomedical applications including imaging and sensing due to their narrow emission peak, size-dependent emission wavelength, and broad excitation range.	Compatibility with biological systems; stable and soluble in biomatrices.	Toxic [135,136].
Graphene quantum dots (GQDs)	Non-viral	Graphene quantum dots (GQDs) are nanoparticles smaller than 100 nm with unique properties, including a low toxicity, chemical stability, and a strong quantum confinement effect. These properties make them promising for use in biological, opto-electronics, energy, and environmental applications.	GQDs advantages include low toxicity, dual functions for gene delivery and nuclear targeting, stable photoluminescence, chemical stability, and pronounced quantum confinement effect.	Further research and clinical investigations are warranted [137,138,139,140,141,142].
Up-conversion nanoparticles (UCNs)	Non-viral	Upconversion nanoparticles (UCNPs) are nanocrystals containing lanthanide ions with optical properties.	Possess a photoactive delivery platform and can be surface modified to enhance their efficiency as gene carriers.	Further research and clinical investigations are warranted [143].
Mesoporous silica nanoparticles (MSNs)	Non-viral	Mesoporous silica nanoparticles (MSNs) have a honeycomb-like, porous structure consisting of silica (SiO_2_).	High surface area, tunable particle and pore sizes, biocompatibility, and easily attachable functional groups. They also have a simple mesoporous or hollow structure, high pore volume, and stimuli-responsive drug release profiles.	Further research and clinical investigations are warranted [144,145,146].
Ferritin	Non-viral	Ferritin forms a naturally occurring nanocage composed of heavy and light polypeptide chains, which can be used as a versatile transport vehicle for various cargo.	Uniform size, high stability in pH and temperature, can be tracked by MRI, and strong humoral immunogenicity.	Further research and clinical investigations are warranted [147,148,149,150].
Red blood cell membrane (RBCM)	Non-viral	RBC membranes have been utilized in drug delivery systems due to their biocompatibility and long half-life. Nanoparticles coated with erythrocyte membranes mimic RBCs and interact with the environment to achieve long-term circulation.	Prolonged circulation, biocompatibility and biodegradability, reduced toxicity, long lifespan, high load capacity, and improved stability and storage.	Further research and clinical investigations are warranted [151].
Gelatin nanoparticles	Non-viral	Gelatin nanoparticles can be hydrolyzed by gelatinases secreted by broad-spectrum bacteria.	Excellent immune evasion, susceptible to hydrolysis by bacteria’s gelatinases into small biomolecules.	Further research and clinical investigations are warranted [151].
DNA nanoclews (DNA NC)	Non-viral	DNA nanoclews are made through rolling circle amplification and are bound to the guide segment of Cas9/sgRNA due to partially complementary sequences. They can be further coated with PEI for endosomal escape.	Precise targeting, consistent size, biodegradability, and efficient endosomal release.	Further research and clinical investigations are warranted [152,153,154].
Arrestin domain-containing protein 1 (ARRDC1)-mediated microvesicles (ARMMs)	Non-viral	ARMMs, or arrestin domain-containing protein 1-mediated microvesicles, can transport various macromolecules, such as p53, RNAs, and CRISPR-Cas9/guide RNA, between cells.	The lipid bilayer membrane protects cargo from degradation and immune system detection.	Further research and clinical investigations are warranted [155,156].
Multifunctional peptides	Non-viral	Multifunctional peptides are promising in nanocarrier design, as they can self-assemble into therapeutic or diagnostic delivery vehicles.	Compatibility with living organisms and ability to target specific sites, endless modification possibilities, and capacity for modular and multiplexed delivery systems.	Further research and clinical investigations are warranted [157].
Cell-penetrating peptide-modified carriers	Non-viral	CPP-modified carriers: utilizing cell-penetrating peptides, small peptide domains with <40 amino acids, facilitates the uptake of various molecular cargo, including DNA, due to their ability to easily cross cell membranes.	Selectively targets specific cell types and traverse cellular membranes.	Further research and clinical investigations are warranted [158,159].
Exosomes (endogenous nanocarriers)	Non-viral	Exosomes (natural nanocarriers) are nanoparticles that can transport RNA and proteins, characterized by their size, surface composition, and lipid and protein content.	Flexibility in donor cell, cargo, targeting peptide, loading technique, and delivery route.	Further research and clinical investigations are warranted [160].
Microneedles (MNs)	Physical	Microneedles (MNs) are small needles that can easily penetrate the skin’s outer layer, making them a promising option for pain-free gene delivery. Different types of MNs, including metal, coated, and dissolving MNs, have shown potential in this regard.	MNs can deliver both low- and high-molecular-weight agents, including nucleic acids; are pain-free and patient-friendly; have the potential for self-administration; and are cost-effective. MNs have been used for transdermal delivery of a variety of agents, including siRNA, small molecular weight drugs, oligonucleotides, DNA, peptides, proteins, and inactivated viruses.	Microneedles are limited in that the application site is restricted to the arms, hands, and abdomen. The duration of application is also critical as some dissolving microneedles dissolve completely in 20 min. While microneedles may not be superior to intradermal injection, their combination with electroporation is more effective than injection alone [161].
Gene gun or biolistic particle delivery system	Physical	The biolistic particle delivery system, also known as the gene gun, is a popular technique for delivering DNA or RNA into cells that are hard to transfect using traditional methods. The method employs microparticles that are ~1 μm in diameter, propelled into tissues via pressurized gas.	Efficient DNA delivery	Low efficiency in transfecting small cells and the potential for causing significant tissue damage [162].
Jet injection	Physical	Jet injection is a needle-free physical method that delivers drugs or vaccines into the skin, subcutaneous or intramuscular tissues using high-pressure microdroplets of liquid. The pressure can reach up to 3–4 bar, with droplet velocities ranging from 100–200 ms^−1^.	Reduces patient discomfort and effective in DNA vaccination.	Further research and clinical investigations are warranted [163].
Electroporation	Physical	Electroporation uses high-voltage pulses to temporarily disrupt cell membranes, allowing for the introduction of molecules such as DNA. The permeabilized state is reversible and can be used for loading cells with various molecules.	Improved gene delivery and expression by 20–1000-fold, decreased cost and time, high efficiency and safety, ability to deliver complex payloads to various tissues.	Optimization needed for new environments, variable efficiency, limited cell survival, non-uniform tissue regeneration, possible tissue harm [164,165,166,167,168].
Ultrasound (Sonoporation)	Physical	Ultrasound waves create pores in cell membrane due to cavitation. Ultrasound-mediated gene transfer, referred to as sonoporation, occurs by the induction of transient membrane permeabilization and has been found to significantly increase the uptake and expression of DNA in cells across many organ systems. In addition, it offers a more favorable safety profile compared to other non-viral delivery methods.	Noninvasive, less tissue damage compared to electroporation, ultrasound is highly accepted in the clinical setting, more efficient than naked DNA injection, systemic injection is possible	Requires careful control of ultrasound parameters to avoid potential side effects [168,169,170,171].
Magnetofection	Physical	Magnetofection is a nanoparticle-mediated approach for transfection of cells, tissues, and tumors. Specific interest is in using superparamagnetic iron oxide nanoparticles (SPIONs) as delivery system of therapeutic genes. Magnetic particles complexed with DNA and an external magnetic field	Magnetofection boosts gene vector efficacy and enables rapid delivery in minutes. It has shown high transduction efficiency in vitro and in vivo, including magnetic field-guided local transfection in the gastrointestinal tract and blood vessels.	Difficulties in localizing it in vivo and cytotoxicity, which may be impacted by particle size and cell entry [168,172,173].
Naked RNA Injection	Physical	Messenger RNA can be conveyed via naked mRNA injection, bypassing the need for carrier molecules.	Naked mRNA offers streamlined preparation, storage, and cost efficiency. Mouse intramuscular injection yields diverse protein expression. Human intradermal injection demonstrates exogenous mRNA translation. Tumor growth inhibition is achieved through intratumoral saline-formulated mRNA encoding four cytokines.	Naked mRNA faces swift degradation by exo- and endonucleases, curtailing protein translation duration. However, PEG-conjugated cationic polymer packaging within polyplex micelles enhances mRNA protection over 10,000-fold compared to naked mRNA, even in ribonuclease-rich serum conditions [174,175,176,177,178].
Naked DNA Injection	Physical	DNA directly injected to site of interest.	It is the most affordable and simple method for delivering DNA, and it can be used to target specific areas of the body for DNA uptake.	Inconsistent and low gene expression, tissue damage around injection area [168].
Gene-Activated Matrix	Physical	Gene-Activated Matrix (GAM)—novel approach that combines gene therapy and tissue engineering to restore damaged tissues.	Targeted and prolonged gene expression, options for both in vivo and ex vivo methods, scaffold for tissue regeneration in three dimensions.	Additional vectors might be necessary to enhance expression, potential DNA damage during matrix construction [168,179].
Hydrodynamics high pressure injection	Physical	Rapid injection of large pDNA volume.	Straightforward, user-friendly, adaptable, and effective.	Further research and clinical investigations are warranted [180,181].

**Table 3 pharmaceuticals-16-01416-t003:** Approved human gene therapy products.

Product	Disease	Ex/In	Vector	Target	Approved by	Date	Mechanism of Action
IMLYGIC (Talimogenelaherparepvec)	Melanoma, pancreatic cancer	In vivo	HSV-1 oncolytic virus	GM-CSF	USA FDA	10/2015	IMLYGIC utilizes a modified herpes simplex virus type 1 (HSV-1) to target cancer cells. The virus has been engineered to have deletions in the y34.5 and a47 regions, and the GM-CSF gene has been inserted into the deleted y34.5 loci. This modification enables the virus to selectively replicate in cancer cells, leading to their destruction and the induction of an antitumor immune response [7,182,183].
LUXTURNA (Voretigene Neparvovec)	Retinal dystrophy (biallelic RPE65 mutation-associated)	In vivo	AAV2	RPE65	USA FDA, EU FDA	12/2017	Luxturna utilizes AAV2 to target RPE cells and introduce a normal copy of the RPE65 gene, which compensates for biallelic mutation. The normal RPE65 protein produced as a result acts as isomerohydrolase, converting trans-retinyl esters to 11-cis-retinal, the natural chromophore of photoreceptor opsins, thereby restoring vision. It is applied via subretinal injection following vitrectomy [7,184,185].
Zolgensma (Onasemnogene Abeparvovec)	Spinal muscular atrophy (Type I)	In vivo	AAV9	SMN1 in motor neurons	USA FDA	05/2019	Zolgensma utilizes a non-replicating recombinant AAV9 vector to deliver a functional copy of the human SMN1 gene to motor neurons in patients with spinal muscular atrophy (SMA). The AAV9 capsid allows for efficient delivery across the blood–brain barrier through intravenous administration, and the modified AAV ITR produces a self-complementary DNA molecule to enhance transcription. The CMV enhancer and chicken-beta-actin-hybrid promoter (CB) control the expression of SMN1 specifically in motor neurons [7,186].
Spinraza (Nusinersen)	Spinal muscular atrophy (Type I)	In vivo	ASO	Intron 7 on the SMN2 hnRNA	USA FDA, EU FDA	12/2016	Nusinersen works by targeting intron 7 on the SMN2 hnRNA through an ASO approach. By increasing inclusion of exon 7 in the final RNA, alternative splicing is modulated, resulting in higher levels of the functional SMN protein in the central nervous system (CNS) [7,187].
Patisiran (Onpattro)	Polyneuropathy (familial amyloidotic polyneuropathy), Hereditary transthyretin-mediated (hATTR) amyloidosis = ATTRv amyloidosis	In vivo	An LNP containing an RNAi	Transthyretin (TTR) gene	EU FDA	08/2018	Patisiran, a siRNA, is designed to attach to and block the genetic material responsible for producing transthyretin. This reduces the production of defective transthyretin, resulting in less amyloid formation and symptom relief [7,188].
Gendicine (rAd-p53)	Carcinoma (head and neck squamous cell carcinoma)	In vivo	Adenovirus	Tp53 tumour-suppressor gene	Chinese State FDA	10/2003	Gendicine uses an adenovirus vector to deliver wild-type p53 into cancer cells. This leads to an increase in p53 production, which can inhibit cell proliferation and induce apoptosis, ultimately helping to treat cancer [7,189,190].
Neovasculgen (PI-VEGF-165)	Atherosclerotic peripheral arterial disease (PAD), including critical limb ischemia (CLI) transmuscular transfer (calf muscles)	In vivo	pDNA	165-amino-acid isoform of human vascular endothelial growth factor (pCMV-VEGF165)	Russian Ministry of Healthcare	12/2011	Neovasculgen utilizes VEGF, an angiogenic effector, to induce angiogenesis and endothelial renovation. The recombinant DNA in Neovasculgen contains a transcription start site, the encoding VEGF165 isoform, a polyadenylation signal, a splicing signal, and SV40 transcription terminator [7,191].
Oncorine (rAd5-H101)	Nasopharyngeal cancer (head and neck cancer, lung cancer, liver cancer, pancreatic cancer)	In vivo	Oncolytic adenovirus type 5	E1B-55 kDa gene-deleted replication-selective adenovirus	Chinese State Food and Drug Administration	11/2005	Oncorine functions by eliminating the E1B-55 KD gene in adenovirus, which is responsible for p53 inactivation. This causes the Oncorine adenovirus to selectively propagate in P53-deficient cancer cells, while normal cells are unaffected. Oncorine-mediated cell cytotoxicity is initiated by the lysis of cancer cells, followed by the release and infection of neighboring cells by adenoviruses [7,192,193].
Defibrotide (Defitelio, defibrotide sodium)	SOS/VOD with multiorgan dysfunction	In vivo	ODN	Vascular endothelial cells	EU FDA	10/2013	Defibrotide’s mechanism of action involves the use of single-stranded oligo DNAs derived from porcine mucosa tissue. These DNAs are designed with aptameric function that can bind to and activate vascular endothelial cells [7,194].
ADSTILADRIN (nadofaragene firadenovec-vncg)	Bacillus Calmette–Guérin (BCG)-unresponsive non-muscle invasive bladder cancer (NMIBC) with carcinoma in situ (CIS) with or without papillary tumors	In vivo	Recombinant adenovirus serotype 5 vector	alfa-2b (IFNα2b)	USA FDA	12/2022	ADSTILADRIN uses a non-replicating adenoviral vector to deliver a copy of the gene that encodes for human interferon-alfa 2b (IFNα2b) to the bladder urothelium. Upon intravesical instillation, ADSTILADRIN causes cell transduction and temporary expression of IFNα2b, which is expected to have anti-tumor properties [195,196].
HEMGENIX (etranacogene dezaparvovec-drlb)	Hemophilia B (congenital factor IX deficiency)	In vivo	Adeno-associated virus serotype 5 (AAV5)	hFIX598 Padua	USA FDA	11/2022	HEMGENIX is based on adeno-associated virus serotype 5 (AAV5), which is designed to deliver a copy of the Padua variant of human coagulation factor IX (hFIX Padua) gene. Upon a single intravenous infusion, HEMGENIX transduces cells and leads to an increase in circulating factor IX activity in patients with hemophilia B [197,198,199].
Upstaza (eladocagene exuparvovec)	Aromatic L-amino acid decarboxylase (AADC) deficiency	In vivo	AAV	AADC	EU FDA	07/2022	Upstaza is designed to address aromatic L-amino acid decarboxylase (AADC) deficiency, a rare genetic disorder caused by mutations in the gene that produces the AADC enzyme. AADC is essential for producing dopamine, a neurotransmitter that controls movement. The therapy is based on an AAV virus carrying a functional copy of the AADC gene that can enter nerve cells, allowing them to produce the missing enzyme and improve dopamine production, leading to symptom relief [200,201].
ABECMA (idecabtagene vicleucel)	Multiple myeloma	Ex vivo	Lentiviral vector (LVV)	A chimeric antigen receptor (CAR)	USA FDA, EU FDA	03/2021	ABECMA uses autologous human T cells, which are genetically modified outside of the body using a lentiviral vector that encodes a chimeric antigen receptor (CAR). This CAR recognizes the B cell maturation antigen (BCMA), which is expressed on the surface of multiple myeloma cells. Once infused back into the patient, the modified T cells can selectively recognize and eliminate the cancer cells expressing BCMA, providing a targeted treatment for multiple myeloma [202,203,204,205].
CARVYKTI (ciltacabtagene autoleucel)	Multiple myeloma	Ex vivo	Lentiviral vector (LVV)	Anti-B cell maturation antigen (BCMA) chimeric antigen receptor (CAR)	USA FDA, EU FDA	02/2022	Carvykti is a gene therapy product containing ciltacabtagene autoleucel, which is composed of the patient’s own T cells that have been genetically modified to express a chimeric antigen receptor (CAR). This CAR is designed to bind to the B-cell maturation antigen (BCMA), a protein present on the surface of multiple myeloma cells, leading to the destruction of these cancerous cells by the modified T cells [206,207,208,209].
SKYSONA (elivaldogene autotemcel)	Cerebral adrenoleukodystrophy (slows the progression of neurologic dysfunction with early, active cerebral adrenoleukodystrophy) (CALD)	Ex vivo	Lenti-D LVV	ABCD1 gene	USA FDA	09/2022 (Accelerated Approval)	SKYSONA utilizes the patient’s own hematopoietic stem cells (HSCs) collected through apheresis procedures. The CD34+ cells are then transduced ex vivo with a replication-incompetent, self-inactivating lentiviral vector (Lenti-D LVV) carrying ABCD1 cDNA, which encodes normal ALDP. The modified viral promoter, internal MNDU3, controls expression of the ABCD1 gene in HSCs and their progeny across all lineages [210,211,212].
YESCARTA (axicabtagene ciloleucel)	Melanoma, pancreatic cancer	Ex vivo	Gamma-retroviral vector	Anti-CD19 CAR gene	USA FDA	10/2017	Yescarta uses autologous T cells transfected with a gamma-retroviral vector expressing a chimeric antigen receptor (CAR) directed against CD19. This CAR consists of an extracellular murine anti-CD19 single-chain variable fragment fused to a cytoplasmic domain composed of CD28 and CD3-zeta co-stimulatory domains. After the autologous T cells are harvested from patients, they are enriched and activated before transduction with the retroviral vector in a closed system in the Yeskarta manufacturing center. The entire manufacturing process takes less than 10 days before the CAR T cells are ready for infusion back into the patient [7,213,214].
KYMRIAH (tisagenlecleucel CTL 019)	Lymphoma (relapsed or refractory follicular)	Ex vivo	Lentiviral	Anti-CD19 chimeric antigen receptor (CAR)	USA FDA, EU FDA	08/2018	Kymriah is an autologous T cell therapy that utilizes a lentiviral delivery system to genetically engineer T cells to express a chimeric antigen receptor (CAR) consisting of a murine single-chain antibody fragment (scFv) that targets CD19, and an intracellular cytoplasmic domain for 4-1BB and CD3 zeta with a CD8 transmembrane hinge. The lentiviral vector is self-inactivating and pseudotyped with a VSV g envelope from the HIV-1 genome, resulting in the integration of the tisagenlecleucel CAR under the control of a constitutively active promoter. Upon binding to CD19-expressing cells, the activated tisagenlecleucel CAR initiates antitumor activity through the CD3 domain, with the intracellular 4-1BB costimulatory domain augmenting the response and promoting long-term persistence of CAR T cells [6,7,166,215,216,217].
Strimvelis (GSK-2696273)	ADA-SCID (severe combined immunodeficiency due to adenosine deaminase deficiency)	Ex vivo	Retroviral	ADA cDNA	EU FDA	05/2016	Strimvelis is used to treat severe combined immunodeficiency due to adenosine deaminase deficiency (ADA-SCID). This rare inherited condition is caused by a mutation in the gene responsible for producing the ADA enzyme, which is essential for maintaining healthy white blood cells. Strimvelis is used in patients who cannot receive a bone marrow transplant due to lack of a suitable, matched donor. The treatment involves genetically modifying the patient’s own bone marrow cells to contain a working ADA gene and then delivering these cells back into the body [7,218].
BREYANZI (lisocabtagene maraleucel)	Large B cell lymphoma (LBCL)	Ex vivo	Lentiviral	CD19 chimeric antigen receptor (CAR)	USA FDA, EU FDA	04/2022	BREYANZI involves the use of a lentiviral vector to genetically modify autologous human T cells. These modified T cells express an anti-CD19 chimeric antigen receptor (CAR) that is made up of CD8+ and CD4+ cell components. The therapy is administered at a strength of 1.1–70 × 106 CAR+ viable T cells/mL for each component [219,220,221].
TECARTUS (brexucabtagene autoleucel)	Acute lymphoblastic leukemia (ALL)	Ex vivo	Retroviral	Anti-CD19 CD28/CD3-zeta chimeric antigen receptor (CAR)	USA FDA, EU FDA	12/2020	TECARTUS uses a patient’s own T cells which are genetically modified ex vivo with a retroviral vector. The vector contains an anti-CD19 chimeric antigen receptor (CAR) comprising a murine anti-CD19 single chain variable fragment (scFv) linked to the CD28 co-stimulatory domain and the CD3-zeta signaling domain [217,222,223,224].
ZYNTEGLO (betibeglogene autotemcel)	Beta-thalassemia	Ex vivo	Lentiglobin BB305 lentiviral vector	Beta-A-T87Q-globin	USA FDA	08/2022	ZYNTEGLO uses autologous CD34+ cells, which contain hematopoietic stem cells that have been transduced with a lentiviral vector called lentiglobin BB305. This vector encodes the beta-A-T87Q-globin gene, which is intended to increase the production of hemoglobin in patients with transfusion-dependent beta-thalassemia [225].
Libmeldy (atidarsagene autotemcel)	Leukodystrophy, Metachromatic	Ex vivo	Lentiviral	ARSA	EU FDA	12/2020	Libmeldy involves extracting CD34+ cells, which are capable of producing white blood cells, from the patient’s blood or bone marrow. Using a modified lentivirus, a gene that enables the production of ARSA is inserted into the CD34+ cells. Once administered back into the patient, the modified cells grow and produce normal white blood cells that can break down sulfatides in surrounding cells, alleviating disease symptoms. The therapy is expected to have long-lasting effects [226].

## Data Availability

Data sharing is not applicable.

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
