# Peer review of "Current State of Human Gene Therapy: Approved Products and Vectors"

_pharmaceuticals, 2023, doi:10.3390/ph16101416_

Round 1
Reviewer 1 Report
I believe that the study has sufficient merit to be considered for publication although major revisions are required.
1. Abstract: In the abstract, the authors could include specific details about the current state of gene therapy, such as the number of approved products and the advancements made in the field. Additionally, it might be beneficial to mention the significance of the various categories of treated diseases.
2. Introduction: it could be restructured to present the historical evolution of gene therapy and the achieved milestones in a more coherent manner. The current approach seems fragmented, and integrating key events, like the approval of Kymriah, more seamlessly could enhance the narrative.
In line 32-33 the authors could clarify which diseases are involved and how exactly gene therapy tackles their root causes.
In line 62-64 they could provide an idea of what approximate percentage of combinations actually succeed and whether there are known examples of successful cases that can support this statement.
In paragraph 2.2 I recommend to the authors this reference that I think is important and that can be of great help when modifying the manuscript (doi: https://doi.org/10.3390/jcm9123853).
3.Conclusion: The conclusion could be broadened to encompass a wider perspective on the future importance of gene therapy and areas where further improvements are possible. This would provide a stronger closing and reinforce the article's main message.
4. Overall, the text could benefit from greater clarity and a more consistent structure. Some passages appear complex and might be simplified to enhance comprehension.
4.References: Ensure that all references and citations are correctly formatted according to the journal.
5.Grammar and Syntax Review:Thoroughly check the article for any grammatical, syntactical, or punctuation errors that could compromise overall clarity.
Moderate editing of English language required
Author Response
Dear Reviewers,
We extend our gratitude for affording us the privilege to present a revised version of the manuscript titled "Current State of Human Gene Therapy: Approved Products and Vectors" for consideration in MDPI Pharmaceuticals. We acknowledge the reviewers' meticulous examination of our work and are indebted for their astute observations, which have significantly enriched our paper. The majority of the suggestions from the reviewers have been thoughtfully integrated, and these modifications are denoted in yellow within the manuscript. We have also provided a comprehensive response to the reviewers' remarks and queries, delineated in blue, in the following section.
REVIEWER 1.
-
Abstract: In the abstract, the authors could include specific details about the current state of gene therapy, such as the number of approved products and the advancements made in the field. Additionally, it might be beneficial to mention the significance of the various categories of treated diseases.
-
Appreciated. The necessary corrections have been made.
-
Line 12-20
Abstract: In the realm of gene therapy, the pivotal moment arrived with Paul Berg's groundbreaking identification of the first recombinant DNA in 1972. This achievement set the stage for future breakthroughs. Conditions once considered insurmountable, like melanoma, pancreatic cancer, and a host of other ailments now are being addressed at their root cause — the genetic level. Presently, the gene therapy landscape stands adorned with 22 approved in-vivo and ex-vivo products including Imlygic, Luxturna, Zolgensma, Spinraza, Patisiran, and many more. In this comprehensive exploration, we delve into a rich assortment of 16 drugs, from siRNA, miRNA, and CRISPR/Cas9 to DNA aptamers and TRAIL/APO2L, as well as 46 carriers, from AAV, AdV, LNPs, and exosomes to naked mRNA, sonoporation, and magnetofection. The article also discusses the advantages and disadvantages of each product and vector type, as well as current challenges faced in the practical use of gene therapy and future potential.
-
Introduction: it could be restructured to present the historical evolution of gene therapy and the achieved milestones in a more coherent manner. The current approach seems fragmented, and integrating key events, like the approval of Kymriah, more seamlessly could enhance the narrative.
-
Appreciated. The necessary corrections have been made.
-
Line 42-68
Delving into the annals of gene therapy's history unveils a narrative shaped by pioneering discoveries and watershed moments. Frederick Griffith's 1928 experiment resonates as a seminal exploration, elucidating the transference of genetic information among bacteria through the transformative process. The monumental breakthrough arrived in 1972 with Paul Berg's revelation of the first recombinant DNA, a milestone that would reverberate across the scientific community. Berg's paradigm-shifting achievement garnered him the 1980 Nobel Prize in Chemistry, underscoring the transformative power of recombining DNA molecules. In a pivotal moment on January 19, 1989, Dr. James A. Wyngaarden, director of the National Institutes of Health (NIH), granted approval for the inaugural clinical protocol, entailing the integration of a foreign gene into immune cells for cancer patients. September 14, 1990, etches another indelible moment as W. French Anderson and his NIH colleagues conducted the first sanctioned gene therapy procedure, a life-altering intervention for a four-year-old afflicted with severe combined immunodeficiency (SCID). From 1990 to 2000, the landscape burgeoned with promise as approximately 300 clinical gene therapy trials embraced around 3000 individuals, marking a dynamic era of exploration. The march of progress ventured eastward as China's State FDA granted approval for Gendicine in 2003 to address Carcinoma, and Oncorine in 2005 for Nasopharyngeal cancer. Russia followed suit, gaining approval for Neovasculgen in 2011 to combat Atherosclerotic Peripheral Arterial Disease (PAD). The historical trajectory surged ahead, with the European FDA approving Defibrotide in 2013 for addressing SOS/VOD with multiorgan dysfunction. In 2015, the USA FDA's imprimatur ushered in a new era with IMLYGIC, ratified to treat Melanoma and Pancreatic Cancer. The progression was resounding, as 2023 bore witness to the culmination of these endeavors, witnessing the approval of 22 gene therapy products by entities such as USA FDA, EU FDA, Chinese State FDA, and the Russian Ministry of Healthcare, emblematic of the multifaceted strides taken to harness the potential of gene therapy on a global scale.
-
In line 32-33 the authors could clarify which diseases are involved and how exactly gene therapy tackles their root causes.
-
Appreciated. The additional content has been included as advised.
-
Line 29-37:
Gene therapy has become a rapidly growing field with significant advancements in recent years. This innovative therapeutic approach is revolutionizing the treatment of various diseases, including melanoma, pancreatic cancer, retinal dystrophy, spinal muscular atrophy, polyneuropathy, hereditary transthyretin-mediated amyloidosis, head and neck squamous cell carcinoma, atherosclerotic peripheral arterial disease, critical limb ischemia, nasopharyngeal cancer, SOS/VOD with multiorgan dysfunction, bacillus calmette-guérin (BCG)-unresponsive non-muscle invasive bladder cancer, hemophilia B, aromatic L-amino acid decarboxylase deficiency, multiple myeloma, cerebral adrenoleukodystrophy, lymphoma, ADA-SCID, large B-cell lymphoma, acute lymphoblastic leukemia, beta-thalassemia, leukodystrophy metachromatic by addressing the root cause of the condition — the genetic level.
-
In line 62-64 they could provide an idea of what approximate percentage of combinations actually succeed and whether there are known examples of successful cases that can support this statement.
-
Appreciated. The additional content has been included as advised.
-
Line 80:
Only about 11,5% of these combinations succeed
Line 1923:
Mullard, A. Parsing clinical success rates. Nat Rev Drug Discov. 2016, 15, 447. https://doi.org/10.1038/nrd.2016.136
-
In paragraph 2.2 I recommend to the authors this reference that I think is important and that can be of great help when modifying the manuscript (doi: https://doi.org/10.3390/jcm9123853).
-
Appreciated. The additional content has been included as advised.
-
Line 122-128:
MicroRNAs (miRNAs) offer a key advantage in gene therapy due to their small size and manipulability. Moreover, around 12 miRNAs have been identified for suppressing endogenous CFTR mRNA expression in the Caco-2 cell line. CFTR, responsible for the monogenic autosomal recessive Cystic Fibrosis (CF), impacts 1 in 3500 global live births. Maria V. Esposito et al. examined 706 CF carriers, revealing undiagnosed CFTR-RD among a subset. Genetic testing scanning analysis aids in CFTR-RD identification, offering potential for tailored follow-up and therapies to enhance outcomes. -
Line 1924-1930
Gillen, A.E.; Gosalia, N.; Leir, S.H.; Harris, A. MicroRNA regulation of expression of the cystic fibrosis transmembrane conductance regulator gene. Biochem J. 2011, 438(1), 25-32. https://10.1042/BJ20110672 -
De Palma, F.D.E.; Raia, V.; Kroemer, G.; Maiuri, M.C. The Multifaceted Roles of MicroRNAs in Cystic Fibrosis. Diagnostics (Basel). 2020, 10(12), 1102. https://10.3390/diagnostics10121102
-
Esposito, M.V.; Aveta, A.; Comegna, M.; Cernera, G.; Iacotucci, P.; Carnovale, V.; Taccetti, G.; Terlizzi, V.; Castaldo, G. Extensive CFTR Gene Analysis Revealed a Higher Occurrence of Cystic Fibrosis Transmembrane Regulator-Related Disorders (CFTR-RD) among CF Carriers. J. Clin. Med. 2020, 9, 3853. https://doi.org/10.3390/jcm9123853
-
Conclusion: The conclusion could be broadened to encompass a wider perspective on the future importance of gene therapy and areas where further improvements are possible. This would provide a stronger closing and reinforce the article's main message.
-
Appreciated. The necessary corrections have been made.
-
Line 1386-1394:
Gene therapy emerges as a robust and versatile tool poised to catalyze the advancement of personalized medicine, revolutionizing therapy efficacy and ameliorating the collateral effects of widely used pharmaceutical agents. A most rapid evolution of personalized medicine is witnessed in the realms of cancer treatment and hereditary disorders, where gene therapy strategies and products play a pivotal role. However, the potential of gene therapy extends far beyond these domains, encompassing a diverse array of pathologies, such as neurodegenerative ailments, immune disorders, inflammatory processes, and more, beckoning for further investigation.
Line 1401-1408:
Moreover, certain ethical, regulatory, and commercialization challenges must be addressed for gene therapy to become more widely adopted. Researchers embarking on the nascent phases of product development are tasked with a multifaceted consideration encompassing current technological frontiers, product commercialization, and the potential for costly solutions. While some of these innovative solutions hold transformative promise, their accessibility for patients could be constrained by elevated costs. It is imperative that these pioneering commercial decisions are underpinned by judicious forethought, as ill-conceived approaches could jeopardize not only production but also the integration of gene therapy products into medical practice. Given the potentially transformative impact of gene therapy, it is critical that these challenges are met with thoughtful and comprehensive solutions.
-
Overall, the text could benefit from greater clarity and a more consistent structure. Some passages appear complex and might be simplified to enhance comprehension.
-
Thank you for your feedback. We sincerely hope that the clarifications and revisions provided during the review process will enhance the content of the article. Given the targeted audience of specialized experts, we believe that the incorporation of more intricate details will not pose challenges to their comprehension of the more complex material. The presentation of more detailed information is anticipated to serve as a valuable asset for their professional endeavors.
-
References: Ensure that all references and citations are correctly formatted according to the journal.
-
Thank you for your diligence in reviewing the references. They have been duly checked and adjusted as needed.
-
Grammar and Syntax Review:Thoroughly check the article for any grammatical, syntactical, or punctuation errors that could compromise overall clarity.
-
Thank you. The article has been reviewed and necessary adjustments have been made to maintain its linguistic precision.
REVIEWER 2.
-
1. Scientifically misleading information was observed. For example, Page 17, Line 488: Onpattro is a lipid nanoparticle (LNP), not a liposome. Please change the description accordingly. The following information will help you to understand the difference between liposomes and lipid nanoparticles. Both liposomes and lipid nanoparticles are lipid-based nanoparticles. Lipid nanoparticles are very similar in the basic physical structure of liposomes. Liposomes are spherical vesicles with an aqueous internal cavity enclosed by a lipid bilayer membrane, whereas lipid nanoparticles do not have aqueous internal cavities. In contrast, lipid nanoparticles form multilayer cores dispersed between lipid layers due to the electrostatic complexation of cationic phospholipids and negatively charged nucleic acid substances.
-
Appreciated. The necessary corrections have been made.
-
Page 38:
Patisiran (Onpattro)
Polyneuropathy (familial amyloidotic polyneuropathy), Hereditary transthyretin-mediated (hATTR) amyloidosis = ATTRv amyloidosis
In-vivo
A liposome containing an RNAi
-
Line 1070:
Patisiran is a liposome containing an RNAi
-
The authors also can introduce micelleplex (J Control Release 2020;323:442-462). Micelleplex is prepared by coating the negatively charged nucleic acids onto positively charged micelle. Indeed, several reports state that the micelleplex outperforms the polyplex regarding colloidal stability and effective delivery of nucleic acid cargoes (Biomacromolecules 2022;23(8):3257-3271).
-
-
Appreciated. The additional content has been included as advised.
-
Page 15-16:
Micelleplex
Non-viral
Micelleplexes, nanostructured NA/micelle-like complexes, draw their unique identity from the distinct chemical composition of amphiphilic copolymers, featuring domains that are cationic and hydrophilic/hydrophobic, thus facilitating the condensation of nucleic acids with one or more cationic blocks.
Effective binding, transportation, and targeted delivery of nucleic acids to cancer cells. The micelleplexes outperform their polyplex counterparts regarding gene silencing, internalization, toxicity, colloidal stability, and payload trafficking.
Nanosystems characterized by excessive positive charge density elicit substantive concerns pertaining to in vivo toxicity, necessitating critical appraisal. [219, 220]
-
Line 637-651:
3.2.12. Micelleplexes -
Micelleplexes, denoting nanostructured NA/micelle-like complexes, are characterized by their distinctive attributes forged by the chemical composition of amphiphilic copolymers. These copolymers showcase regions that are both cationic and hydrophilic/hydrophobic, thereby facilitating the compaction of nucleic acids through interaction with one or more cationic constituents.
-
The virtues of Micelleplexes extend to proficient binding, conveyance, and precision delivery of nucleic acids to neoplastic cells. These micelleplexes notably outperform their polyplex counterparts in the domains of gene silencing, cellular internalization, toxicity mitigation, colloidal stability enhancement, and payload trafficking efficiency.
-
Nevertheless, the progressive landscape of innovation is attended by a dualistic complexion, where nanosystems characterized by an excessive positive charge density beckon forth discerning considerations concerning their in vivo toxicity potential, thereby warranting scrupulous evaluation. [219, 220]
-
Line 1883-1888:
Pereira-Silva, M.; Jarak, I.; Alvarez-Lorenzo, C.; Concheiro, A.; Santos, A.C.; Maleki, R.; Veiga, F.; Figueiras, A. Micelleplexes as nucleic acid delivery systems for cancer-targeted therapies. J Control Release 2020, 323, 442-462. https://doi.org/10.1016/j.jconrel.2020.04.041 -
Grimme, C.J.; Hanson, M.G.; Corcoran, L.G.; Reineke, T.M. Polycation Architecture Affects Complexation and Delivery of Short Antisense Oligonucleotides: Micelleplexes Outperform Polyplexes. Biomacromolecules 2022, 23(8), 3257-3271. https://doi.org/10.1021/acs.biomac.2c00338
-
-
We recommend that the authors introduce the “naked mRNA” section and add the following information. Wolff et al. showed for the first time that intramuscular injection of naked mRNA could lead to the expression of different proteins in mouse muscle for the first time (Science 1990;247(4949 Pt 1):1465-8). The first intradermal injection of naked mRNA into the human dermis showed translation of the exogenous mRNA (Gene Ther 2007;14(15):1175-80). The same study also provided the first insight into the uptake mechanism of naked mRNA. Intratumoral administration of saline-formulated mRNA encoding four cytokines (interleukin-12 (IL-12) single chain, interferon-⍺, granulocyte-macrophage colony-stimulating factor, and IL-15 sushi) effectively inhibited tumor growth (Sci Transl Med 202;13(610):eabc7804).
The use of delivery carriers for gene therapeutics is not well established in the manuscript. We recommend that the authors elaborate on the importance of delivery carriers. For example, naked mRNA is subjected to rapid degradation in the extracellular and intracellular spaces by exo- and endonucleases, dramatically decreasing the duration of protein translation from mRNA. Packaging mRNA using PEG-conjugated cationic polymers within the core of polyplex micelle substantially protected the remaining intact mRNA by more than 10000-fold compared to naked mRNA upon ribonuclease-rich serum incubation (J Drug Target 2019;27(5-6):670-680).-
Appreciated. The additional content has been included as advised.
-
-
Page 19:
|
Naked RNA Injection |
Physical |
Messenger RNA can be conveyed via naked mRNA injection, bypassing the need for carrier molecules. |
Naked mRNA offers streamlined preparation, storage, and cost efficiency. Mouse intramuscular injection yields diverse protein expression. Human intradermal injection demonstrates exogenous mRNA translation. Tumor growth inhibition is achieved through intratumoral saline-formulated mRNA encoding four cytokines. |
Naked mRNA faces swift degradation by exo- and endonucleases, curtailing protein translation duration. However, PEG-conjugated cationic polymer packaging within polyplex micelles enhances mRNA protection over 10000-fold compared to naked mRNA, even in ribonuclease-rich serum conditions [221-225]. |
-
Line 941-960:
3.3.7. Naked RNA injection -
Naked mRNA offers an innovative approach to deliver genetic material without the need for traditional carrier molecules. This strategy entails the direct administration of mRNA through a process known as naked mRNA injection.
-
The allure of this approach lies in its simplicity, efficient preparation, cost-effectiveness, and facile storage. Notably, the technique has demonstrated its versatility through intramuscular injection in murine models, yielding diverse protein expression profiles. Similarly, human studies employing intradermal injection have showcased the successful translation of exogenous mRNA, underscoring its potential for human applications. Beyond these achievements, the intratumoral delivery of saline-formulated mRNA encoding four cytokines has emerged as a potent tool, effectively impeding tumor growth.
-
However, despite these strides, the realm of naked mRNA is not devoid of challenges. Swift degradation within extracellular and intracellular environments by exo- and endonucleases poses a notable hurdle, significantly truncating the duration of protein translation stemming from the mRNA. Addressing this issue, innovative strategies such as packaging mRNA within the core of polyplex micelles, employing PEG-conjugated cationic polymers, have arisen. Remarkably, this approach confers substantial protection, shielding intact mRNA from degradation by over 10000-fold compared to naked mRNA, even in conditions rich with ribonucleases [221-225].
-
Line 1889-1904:
Ramachandran, S.; Satapathy, S.R.; Dutta, T. Delivery Strategies for mRNA Vaccines. Pharmaceut Med. 2022, 36(1), 11-20. https://doi.org/10.1007/s40290-021-00417-5 -
Wolff, J.A.; Malone, R.W.; Williams, P.; Chong, W.; Acsadi, G.; Jani, A.; Felgner, P.L. Direct gene transfer into mouse muscle in vivo. Science 1990, 247(4949 Pt 1), 1465-8. https://doi.org/10.1126/science.1690918
-
Probst, J.; Weide, B.; Scheel, B.; Pichler, B.J.; Hoerr, I.; Rammensee, H.G.; Pascolo, S. Spontaneous cellular uptake of exogenous messenger RNA in vivo is nucleic acid-specific, saturable and ion dependent. Gene Ther. 2007, 14(15), 1175-80. https://doi.org/10.1038/sj.gt.3302964
-
Hotz, C.; Wagenaar, T.R.; Gieseke, F.; Bangari, D.S.; Callahan, M.; Cao, H.; Diekmann, J.; Diken, M.; Grunwitz, C.; Hebert, A.; Hsu, K.; Bernardo, M.; Karikó, K.; Kreiter, S.; Kuhn, A.N.; Levit, M.; Malkova, N.; Masciari, S.; Pollard, J.; Qu, H.; Ryan, S.; Selmi, A.; Schlereth, J.; Singh, K.; Sun, F.; Tillmann, B.; Tolstykh, T.; Weber, W.; Wicke, L.; Witzel, S.; Yu, Q.; Zhang, Y.A.; Zheng, G.; Lager, J.; Nabel, G.J.; Sahin, U.; Wiederschain, D. Local delivery of mRNA-encoded cytokines promotes antitumor immunity and tumor eradication across multiple preclinical tumor models. Sci Transl Med. 2021, 13(610), eabc7804. https://doi.org/10.1126/scitranslmed.abc7804
-
Dirisala, A.; Uchida, S.; Tockary, T.A.; Yoshinaga, N.; Li, J.; Osawa, S.; Gorantla, L.; Fukushima, S.; Osada, K.; Kataoka, K. Precise tuning of disulphide crosslinking in mRNA polyplex micelles for optimising extracellular and intracellular nuclease tolerability. J Drug Target. 2019, 27(5-6), 670-680. https://doi.org/10.1080/1061186X.2018.1550646
-
Future perspectives may be essential to provide the author’s suggestions to the research community. For example, Nonspecific sequestration (cell surface binding and endocytic uptake) of gene therapeutics-loaded delivery carriers by the reticuloendothelial system (RES) organs, mainly the liver sinusoidal scavenger wall cells (liver resident macrophagic Kupffer cells + liver sinusoidal endothelial cells), is the biggest hurdle to the clinical translation (PLoS Pathog2011;7(9):e1002281. This sequestration not only substantially decreases the delivery of a sufficient dose to the target site but also often can raise toxicity and immunogenicity issues. Transient and selective blockading of the scavenging functions of liver sinusoidal wall cells by stealth coating with PEG-oligopeptides prevented the sinusoidal capture of nonviral and viral gene vectors, thereby boosting their gene transfer efficiency in the target tissues (Sci Adv 2020;6(26):eabb8133). Undesired off-target accumulation of current FDA-approved mRNA-encapsulated LNP to the liver, even after local intramuscular administration (Nat Biomed Eng 2019;3(5): 371-380), induces adverse side effects in the liver, such as liver cell necrosis and T cell-mediated hepatitis, possibly due to the unintended expression of antigens in the liver, which limits their widespread applications (Proc Natl Acad Sci U S A 2022;119(34): e2207841119). In addition, such hepatic off-targeting exacerbates the pre-existing inflammation in the liver (J Control Release 2022;344:50-61). Minimizing hepatic off-targeting while maximizing the delivery at the injection site could be advantageous for not only widespread vaccine applications. Furthermore, such magic bullet-like targeting at the diseased tissue with minimal hepatic targeting could be helpful for patients with preexisting systemic inflammation (https://doi.org/10.3389/fbioe.2023.1274210).
-
Appreciated. The additional content has been included as advised.
-
Line 1344-1369:
5. Future Directions -
As we cast our gaze towards the horizon, the landscape of gene therapy reveals a realm brimming with potentialities yet to be harnessed. With a deepening comprehension of the intricacies of the human genome and the maturation of our genetic manipulation tools, the vista of gene therapy seems poised for continued expansion. The path ahead is one of unwavering research and innovative strides, a journey that holds the promise of ushering forth more potent treatments and transformative remedies for a diverse spectrum of ailments. The transformative impact on the lives of patients globally is on the precipice of realization. For instance, a paramount obstacle on the path to clinical translation resides in the nonspecific sequestration of gene therapeutics-loaded delivery carriers by the reticuloendothelial system (RES) organs, notably the liver sinusoidal scavenger wall cells. This not only compromises targeted delivery but also accentuates concerns of toxicity and immunogenicity. Stealth coating with PEG-oligopeptides emerges as a strategic intervention, transiently and selectively obstructing the scavenging functions of liver sinusoidal wall cells. This breakthrough approach circumvents sinusoidal capture, thus elevating gene transfer efficacy within target tissues. Moreover, the lingering challenge of undesired hepatic off-target accumulation of FDA-approved mRNA-encapsulated LNP persists, even following local intramuscular administration. This inadvertently triggers adverse hepatic side effects and inflammation, constraining their broad application. A paradigm shift beckons, characterized by minimizing hepatic off-targeting while optimizing injection site delivery. Such precision targeting not only augments vaccine applications but also holds the promise of benefiting patients beset by systemic inflammation. The confluence of innovative solutions with the pursuit of precision heralds the dawn of an era where gene therapy's potential reaches unprecedented heights [226-231].
-
-
Line 1905-1922:
Ganesan, L.P.; Mohanty, S.; Kim, J.; Clark, K.R.; Robinson, J.M.; Anderson, C.L. Rapid and Efficient Clearance of Blood-borne Virus by Liver Sinusoidal Endothelium. PLoS Pathog. 2011, 7(9), e1002281. https://doi.org/10.1371/journal.ppat.1002281 -
Dirisala, A.; Uchida, S.; Toh, K.; Li, J.; Osawa, S.; Tockary, T.A.; Liu, X.; Abbasi, S.; Hayashi, K.; Mochida, Y.; Fukushima, S.; Kinoh, H.; Osada, K.; Kataoka, K. Transient stealth coating of liver sinusoidal wall by anchoring two-armed PEG for retargeting nanomedicines. Sci. Adv. 2020, 6, eabb8133. https://doi.org/10.1126/sciadv.abb8133
-
Lindsay, K.E.; Bhosle, S.M.; Zurla, C.; Jared Beyersdorf; Kenneth A. Rogers; Daryll Vanover; Peng Xiao; Mariluz Araínga; Lisa M. Shirreff; Bruno Pitard; Patrick Baumhof; Francois Villinger; Philip J. Santangelo. Visualization of early events in mRNA vaccine delivery in non-human primates via PET–CT and near-infrared imaging. Nat. Biomed. Eng. 2019, 3, 371–380. https://doi.org/10.1038/s41551-019-0378-3
-
Chen, J.; Ye, Z.; Huang, C.; Qiu, M.; Song, D.; Li, Y.; Xu, Q. Lipid nanoparticle-mediated lymph node-targeting delivery of mRNA cancer vaccine elicits robust CD8+ T cell response. Proc Natl Acad Sci U S A. 2022, 23, 119(34), e2207841119. https://doi.org/10.1073/pnas.2207841119
-
Parhiz, H.; Brenner, J.S.; Patel, P.N.; Papp, T.E.; Shahnawaz, H.; Li, Q.; Shi, R.; Zamora, M.E.; Yadegari, A.; Marcos-Contreras, O.A.; Natesan, A.; Pardi, N.; Shuvaev, V.V.; Kiseleva, R.; Myerson, J.W.; Uhler, T.; Riley, R.S.; Han, X.; Mitchell, M.J.; Lam, K.; Heyes, J.; Weissman, D.; Muzykantov, V.R. Added to pre-existing inflammation, mRNA-lipid nanoparticles induce inflammation exacerbation (IE). J Control Release. 2022, 344, 50-61. https://doi.org/10.1016/j.jconrel.2021.12.027
-
Dirisala, A.; Li, J.; Gonzalez-Carter, D.; Wang, Z. Editorial: Delivery systems in biologics-based therapeutics. Front. Bioeng. Biotechnol. 2023, 11, 1274210. https://doi.org/10.3389/fbioe.2023.1274210
-
-
Expand the full forms of abbreviations or acronyms. For example, Page 6, Line 173: CRISPR/Cas9 --> clustered regularly interspersed short palindromic repeats (CRISPR)/CRISPR-associated protein 9 (Cas9)
-
Appreciated. The necessary corrections have been made.
-
Redundant abbreviations or acronyms were observed. They will increase the word count but not the scientific information. Abbreviate or acronym after their first appearance in the manuscript. For example: Lines 187, 188, and 651: Plasmid DNA (pDNA) Lines 154, and 155:Oligodeoxynucleotides (ODNs) Table 2 and Line 896: superparamagnetic iron oxide nanoparticles (SPIONs)
-
Appreciated. The necessary corrections have been made.
We anticipate that the refined manuscript aligns more closely with the standards of MDPI Pharmaceuticals, yet remain receptive to any additional feedback that may arise. We extend our gratitude for your attentive consideration and valued insights.
Awaiting your response to our revisions, we remain.
Best regards,
On behalf of the team of authors,
Aladdin Y. Shchaslyvyi

Reviewer 2 Report
Aladdin Y. Shchaslyvyi and team reviewed a topic entitled “Current State of Human Gene Therapy: Approved Products and Vectors”. The authors discussed the current status of gene therapy and tabulated approved gene therapeutics. Moreover, the team also focused on the advantages and disadvantages of various gene delivery vectors, ranging from nature-derived viruses to synthetically engineered non-viral vectors. Overall, the manuscript is attractive to researchers of nucleic acid delivery and is worth publishing in MDPI Pharmaceuticals. However, we recommend that the authors consider the following suggestions to reach out to broad audiences of different disciplines.
1. Scientifically misleading information was observed. For example, Page 17, Line 488: Onpattro is a lipid nanoparticle (LNP), not a liposome. Please change the description accordingly. The following information will help you to understand the difference between liposomes and lipid nanoparticles. Both liposomes and lipid nanoparticles are lipid-based nanoparticles. Lipid nanoparticles are very similar in the basic physical structure of liposomes. Liposomes are spherical vesicles with an aqueous internal cavity enclosed by a lipid bilayer membrane, whereas lipid nanoparticles do not have aqueous internal cavities. In contrast, lipid nanoparticles form multilayer cores dispersed between lipid layers due to the electrostatic complexation of cationic phospholipids and negatively charged nucleic acid substances.
2. The authors also can introduce micelleplex (J Control Release 2020;323:442-462). Micelleplex is prepared by coating the negatively charged nucleic acids onto positively charged micelle. Indeed, several reports state that the micelleplex outperforms the polyplex regarding colloidal stability and effective delivery of nucleic acid cargoes (Biomacromolecules 2022;23(8):3257-3271).
3. We recommend that the authors introduce the “naked mRNA” section and add the following information. Wolff et al. showed for the first time that intramuscular injection of naked mRNA could lead to the expression of different proteins in mouse muscle for the first time (Science 1990;247(4949 Pt 1):1465-8). The first intradermal injection of naked mRNA into the human dermis showed translation of the exogenous mRNA (Gene Ther 2007;14(15):1175-80). The same study also provided the first insight into the uptake mechanism of naked mRNA. Intratumoral administration of saline-formulated mRNA encoding four cytokines (interleukin-12 (IL-12) single chain, interferon-⍺, granulocyte-macrophage colony-stimulating factor, and IL-15 sushi) effectively inhibited tumor growth (Sci Transl Med 202;13(610):eabc7804).
4. The use of delivery carriers for gene therapeutics is not well established in the manuscript. We recommend that the authors elaborate on the importance of delivery carriers. For example, naked mRNA is subjected to rapid degradation in the extracellular and intracellular spaces by exo- and endonucleases, dramatically decreasing the duration of protein translation from mRNA. Packaging mRNA using PEG-conjugated cationic polymers within the core of polyplex micelle substantially protected the remaining intact mRNA by more than 10000-fold compared to naked mRNA upon ribonuclease-rich serum incubation (J Drug Target 2019;27(5-6):670-680).
5. Future perspectives may be essential to provide the author’s suggestions to the research community. For example, Nonspecific sequestration (cell surface binding and endocytic uptake) of gene therapeutics-loaded delivery carriers by the reticuloendothelial system (RES) organs, mainly the liver sinusoidal scavenger wall cells (liver resident macrophagic Kupffer cells + liver sinusoidal endothelial cells), is the biggest hurdle to the clinical translation (PLoS Pathog2011;7(9):e1002281. This sequestration not only substantially decreases the delivery of a sufficient dose to the target site but also often can raise toxicity and immunogenicity issues. Transient and selective blockading of the scavenging functions of liver sinusoidal wall cells by stealth coating with PEG-oligopeptides prevented the sinusoidal capture of nonviral and viral gene vectors, thereby boosting their gene transfer efficiency in the target tissues (Sci Adv 2020;6(26):eabb8133). Undesired off-target accumulation of current FDA-approved mRNA-encapsulated LNP to the liver, even after local intramuscular administration (Nat Biomed Eng 2019;3(5): 371-380), induces adverse side effects in the liver, such as liver cell necrosis and T cell-mediated hepatitis, possibly due to the unintended expression of antigens in the liver, which limits their widespread applications (Proc Natl Acad Sci U S A 2022;119(34): e2207841119). In addition, such hepatic off-targeting exacerbates the pre-existing inflammation in the liver (J Control Release 2022;344:50-61). Minimizing hepatic off-targeting while maximizing the delivery at the injection site could be advantageous for not only widespread vaccine applications. Furthermore, such magic bullet-like targeting at the diseased tissue with minimal hepatic targeting could be helpful for patients with preexisting systemic inflammation (https://doi.org/10.3389/fbioe.2023.1274210).
6. Expand the full forms of abbreviations or acronyms. For example, Page 6, Line 173: CRISPR/Cas9 --> clustered regularly interspersed short palindromic repeats (CRISPR)/CRISPR-associated protein 9 (Cas9)
7. Redundant abbreviations or acronyms were observed. They will increase the word count but not the scientific information. Abbreviate or acronym after their first appearance in the manuscript. For example: Lines 187, 188, and 651: Plasmid DNA (pDNA) Lines 154, and 155:Oligodeoxynucleotides (ODNs) Table 2 and Line 896: superparamagnetic iron oxide nanoparticles (SPIONs)
Author Response
Dear Reviewers,
We extend our gratitude for affording us the privilege to present a revised version of the manuscript titled "Current State of Human Gene Therapy: Approved Products and Vectors" for consideration in MDPI Pharmaceuticals. We acknowledge the reviewers' meticulous examination of our work and are indebted for their astute observations, which have significantly enriched our paper. The majority of the suggestions from the reviewers have been thoughtfully integrated, and these modifications are denoted in yellow within the manuscript. We have also provided a comprehensive response to the reviewers' remarks and queries, delineated in blue, in the following section.
REVIEWER 2.
-
1. Scientifically misleading information was observed. For example, Page 17, Line 488: Onpattro is a lipid nanoparticle (LNP), not a liposome. Please change the description accordingly. The following information will help you to understand the difference between liposomes and lipid nanoparticles. Both liposomes and lipid nanoparticles are lipid-based nanoparticles. Lipid nanoparticles are very similar in the basic physical structure of liposomes. Liposomes are spherical vesicles with an aqueous internal cavity enclosed by a lipid bilayer membrane, whereas lipid nanoparticles do not have aqueous internal cavities. In contrast, lipid nanoparticles form multilayer cores dispersed between lipid layers due to the electrostatic complexation of cationic phospholipids and negatively charged nucleic acid substances.
-
Appreciated. The necessary corrections have been made.
-
Page 38:
Patisiran (Onpattro)
Polyneuropathy (familial amyloidotic polyneuropathy), Hereditary transthyretin-mediated (hATTR) amyloidosis = ATTRv amyloidosis
In-vivo
A liposome containing an RNAi
-
Line 1070:
Patisiran is a liposome containing an RNAi
-
The authors also can introduce micelleplex (J Control Release 2020;323:442-462). Micelleplex is prepared by coating the negatively charged nucleic acids onto positively charged micelle. Indeed, several reports state that the micelleplex outperforms the polyplex regarding colloidal stability and effective delivery of nucleic acid cargoes (Biomacromolecules 2022;23(8):3257-3271).
-
-
Appreciated. The additional content has been included as advised.
-
Page 15-16:
Micelleplex
Non-viral
Micelleplexes, nanostructured NA/micelle-like complexes, draw their unique identity from the distinct chemical composition of amphiphilic copolymers, featuring domains that are cationic and hydrophilic/hydrophobic, thus facilitating the condensation of nucleic acids with one or more cationic blocks.
Effective binding, transportation, and targeted delivery of nucleic acids to cancer cells. The micelleplexes outperform their polyplex counterparts regarding gene silencing, internalization, toxicity, colloidal stability, and payload trafficking.
Nanosystems characterized by excessive positive charge density elicit substantive concerns pertaining to in vivo toxicity, necessitating critical appraisal. [219, 220]
-
Line 637-651:
3.2.12. Micelleplexes -
Micelleplexes, denoting nanostructured NA/micelle-like complexes, are characterized by their distinctive attributes forged by the chemical composition of amphiphilic copolymers. These copolymers showcase regions that are both cationic and hydrophilic/hydrophobic, thereby facilitating the compaction of nucleic acids through interaction with one or more cationic constituents.
-
The virtues of Micelleplexes extend to proficient binding, conveyance, and precision delivery of nucleic acids to neoplastic cells. These micelleplexes notably outperform their polyplex counterparts in the domains of gene silencing, cellular internalization, toxicity mitigation, colloidal stability enhancement, and payload trafficking efficiency.
-
Nevertheless, the progressive landscape of innovation is attended by a dualistic complexion, where nanosystems characterized by an excessive positive charge density beckon forth discerning considerations concerning their in vivo toxicity potential, thereby warranting scrupulous evaluation. [219, 220]
-
Line 1883-1888:
Pereira-Silva, M.; Jarak, I.; Alvarez-Lorenzo, C.; Concheiro, A.; Santos, A.C.; Maleki, R.; Veiga, F.; Figueiras, A. Micelleplexes as nucleic acid delivery systems for cancer-targeted therapies. J Control Release 2020, 323, 442-462. https://doi.org/10.1016/j.jconrel.2020.04.041 -
Grimme, C.J.; Hanson, M.G.; Corcoran, L.G.; Reineke, T.M. Polycation Architecture Affects Complexation and Delivery of Short Antisense Oligonucleotides: Micelleplexes Outperform Polyplexes. Biomacromolecules 2022, 23(8), 3257-3271. https://doi.org/10.1021/acs.biomac.2c00338
-
-
We recommend that the authors introduce the “naked mRNA” section and add the following information. Wolff et al. showed for the first time that intramuscular injection of naked mRNA could lead to the expression of different proteins in mouse muscle for the first time (Science 1990;247(4949 Pt 1):1465-8). The first intradermal injection of naked mRNA into the human dermis showed translation of the exogenous mRNA (Gene Ther 2007;14(15):1175-80). The same study also provided the first insight into the uptake mechanism of naked mRNA. Intratumoral administration of saline-formulated mRNA encoding four cytokines (interleukin-12 (IL-12) single chain, interferon-⍺, granulocyte-macrophage colony-stimulating factor, and IL-15 sushi) effectively inhibited tumor growth (Sci Transl Med 202;13(610):eabc7804).
The use of delivery carriers for gene therapeutics is not well established in the manuscript. We recommend that the authors elaborate on the importance of delivery carriers. For example, naked mRNA is subjected to rapid degradation in the extracellular and intracellular spaces by exo- and endonucleases, dramatically decreasing the duration of protein translation from mRNA. Packaging mRNA using PEG-conjugated cationic polymers within the core of polyplex micelle substantially protected the remaining intact mRNA by more than 10000-fold compared to naked mRNA upon ribonuclease-rich serum incubation (J Drug Target 2019;27(5-6):670-680).-
Appreciated. The additional content has been included as advised.
-
-
Page 19:
|
Naked RNA Injection |
Physical |
Messenger RNA can be conveyed via naked mRNA injection, bypassing the need for carrier molecules. |
Naked mRNA offers streamlined preparation, storage, and cost efficiency. Mouse intramuscular injection yields diverse protein expression. Human intradermal injection demonstrates exogenous mRNA translation. Tumor growth inhibition is achieved through intratumoral saline-formulated mRNA encoding four cytokines. |
Naked mRNA faces swift degradation by exo- and endonucleases, curtailing protein translation duration. However, PEG-conjugated cationic polymer packaging within polyplex micelles enhances mRNA protection over 10000-fold compared to naked mRNA, even in ribonuclease-rich serum conditions [221-225]. |
-
Line 941-960:
3.3.7. Naked RNA injection -
Naked mRNA offers an innovative approach to deliver genetic material without the need for traditional carrier molecules. This strategy entails the direct administration of mRNA through a process known as naked mRNA injection.
-
The allure of this approach lies in its simplicity, efficient preparation, cost-effectiveness, and facile storage. Notably, the technique has demonstrated its versatility through intramuscular injection in murine models, yielding diverse protein expression profiles. Similarly, human studies employing intradermal injection have showcased the successful translation of exogenous mRNA, underscoring its potential for human applications. Beyond these achievements, the intratumoral delivery of saline-formulated mRNA encoding four cytokines has emerged as a potent tool, effectively impeding tumor growth.
-
However, despite these strides, the realm of naked mRNA is not devoid of challenges. Swift degradation within extracellular and intracellular environments by exo- and endonucleases poses a notable hurdle, significantly truncating the duration of protein translation stemming from the mRNA. Addressing this issue, innovative strategies such as packaging mRNA within the core of polyplex micelles, employing PEG-conjugated cationic polymers, have arisen. Remarkably, this approach confers substantial protection, shielding intact mRNA from degradation by over 10000-fold compared to naked mRNA, even in conditions rich with ribonucleases [221-225].
-
Line 1889-1904:
Ramachandran, S.; Satapathy, S.R.; Dutta, T. Delivery Strategies for mRNA Vaccines. Pharmaceut Med. 2022, 36(1), 11-20. https://doi.org/10.1007/s40290-021-00417-5 -
Wolff, J.A.; Malone, R.W.; Williams, P.; Chong, W.; Acsadi, G.; Jani, A.; Felgner, P.L. Direct gene transfer into mouse muscle in vivo. Science 1990, 247(4949 Pt 1), 1465-8. https://doi.org/10.1126/science.1690918
-
Probst, J.; Weide, B.; Scheel, B.; Pichler, B.J.; Hoerr, I.; Rammensee, H.G.; Pascolo, S. Spontaneous cellular uptake of exogenous messenger RNA in vivo is nucleic acid-specific, saturable and ion dependent. Gene Ther. 2007, 14(15), 1175-80. https://doi.org/10.1038/sj.gt.3302964
-
Hotz, C.; Wagenaar, T.R.; Gieseke, F.; Bangari, D.S.; Callahan, M.; Cao, H.; Diekmann, J.; Diken, M.; Grunwitz, C.; Hebert, A.; Hsu, K.; Bernardo, M.; Karikó, K.; Kreiter, S.; Kuhn, A.N.; Levit, M.; Malkova, N.; Masciari, S.; Pollard, J.; Qu, H.; Ryan, S.; Selmi, A.; Schlereth, J.; Singh, K.; Sun, F.; Tillmann, B.; Tolstykh, T.; Weber, W.; Wicke, L.; Witzel, S.; Yu, Q.; Zhang, Y.A.; Zheng, G.; Lager, J.; Nabel, G.J.; Sahin, U.; Wiederschain, D. Local delivery of mRNA-encoded cytokines promotes antitumor immunity and tumor eradication across multiple preclinical tumor models. Sci Transl Med. 2021, 13(610), eabc7804. https://doi.org/10.1126/scitranslmed.abc7804
-
Dirisala, A.; Uchida, S.; Tockary, T.A.; Yoshinaga, N.; Li, J.; Osawa, S.; Gorantla, L.; Fukushima, S.; Osada, K.; Kataoka, K. Precise tuning of disulphide crosslinking in mRNA polyplex micelles for optimising extracellular and intracellular nuclease tolerability. J Drug Target. 2019, 27(5-6), 670-680. https://doi.org/10.1080/1061186X.2018.1550646
-
Future perspectives may be essential to provide the author’s suggestions to the research community. For example, Nonspecific sequestration (cell surface binding and endocytic uptake) of gene therapeutics-loaded delivery carriers by the reticuloendothelial system (RES) organs, mainly the liver sinusoidal scavenger wall cells (liver resident macrophagic Kupffer cells + liver sinusoidal endothelial cells), is the biggest hurdle to the clinical translation (PLoS Pathog2011;7(9):e1002281. This sequestration not only substantially decreases the delivery of a sufficient dose to the target site but also often can raise toxicity and immunogenicity issues. Transient and selective blockading of the scavenging functions of liver sinusoidal wall cells by stealth coating with PEG-oligopeptides prevented the sinusoidal capture of nonviral and viral gene vectors, thereby boosting their gene transfer efficiency in the target tissues (Sci Adv 2020;6(26):eabb8133). Undesired off-target accumulation of current FDA-approved mRNA-encapsulated LNP to the liver, even after local intramuscular administration (Nat Biomed Eng 2019;3(5): 371-380), induces adverse side effects in the liver, such as liver cell necrosis and T cell-mediated hepatitis, possibly due to the unintended expression of antigens in the liver, which limits their widespread applications (Proc Natl Acad Sci U S A 2022;119(34): e2207841119). In addition, such hepatic off-targeting exacerbates the pre-existing inflammation in the liver (J Control Release 2022;344:50-61). Minimizing hepatic off-targeting while maximizing the delivery at the injection site could be advantageous for not only widespread vaccine applications. Furthermore, such magic bullet-like targeting at the diseased tissue with minimal hepatic targeting could be helpful for patients with preexisting systemic inflammation (https://doi.org/10.3389/fbioe.2023.1274210).
-
Appreciated. The additional content has been included as advised.
-
Line 1344-1369:
5. Future Directions -
As we cast our gaze towards the horizon, the landscape of gene therapy reveals a realm brimming with potentialities yet to be harnessed. With a deepening comprehension of the intricacies of the human genome and the maturation of our genetic manipulation tools, the vista of gene therapy seems poised for continued expansion. The path ahead is one of unwavering research and innovative strides, a journey that holds the promise of ushering forth more potent treatments and transformative remedies for a diverse spectrum of ailments. The transformative impact on the lives of patients globally is on the precipice of realization. For instance, a paramount obstacle on the path to clinical translation resides in the nonspecific sequestration of gene therapeutics-loaded delivery carriers by the reticuloendothelial system (RES) organs, notably the liver sinusoidal scavenger wall cells. This not only compromises targeted delivery but also accentuates concerns of toxicity and immunogenicity. Stealth coating with PEG-oligopeptides emerges as a strategic intervention, transiently and selectively obstructing the scavenging functions of liver sinusoidal wall cells. This breakthrough approach circumvents sinusoidal capture, thus elevating gene transfer efficacy within target tissues. Moreover, the lingering challenge of undesired hepatic off-target accumulation of FDA-approved mRNA-encapsulated LNP persists, even following local intramuscular administration. This inadvertently triggers adverse hepatic side effects and inflammation, constraining their broad application. A paradigm shift beckons, characterized by minimizing hepatic off-targeting while optimizing injection site delivery. Such precision targeting not only augments vaccine applications but also holds the promise of benefiting patients beset by systemic inflammation. The confluence of innovative solutions with the pursuit of precision heralds the dawn of an era where gene therapy's potential reaches unprecedented heights [226-231].
-
-
Line 1905-1922:
Ganesan, L.P.; Mohanty, S.; Kim, J.; Clark, K.R.; Robinson, J.M.; Anderson, C.L. Rapid and Efficient Clearance of Blood-borne Virus by Liver Sinusoidal Endothelium. PLoS Pathog. 2011, 7(9), e1002281. https://doi.org/10.1371/journal.ppat.1002281 -
Dirisala, A.; Uchida, S.; Toh, K.; Li, J.; Osawa, S.; Tockary, T.A.; Liu, X.; Abbasi, S.; Hayashi, K.; Mochida, Y.; Fukushima, S.; Kinoh, H.; Osada, K.; Kataoka, K. Transient stealth coating of liver sinusoidal wall by anchoring two-armed PEG for retargeting nanomedicines. Sci. Adv. 2020, 6, eabb8133. https://doi.org/10.1126/sciadv.abb8133
-
Lindsay, K.E.; Bhosle, S.M.; Zurla, C.; Jared Beyersdorf; Kenneth A. Rogers; Daryll Vanover; Peng Xiao; Mariluz Araínga; Lisa M. Shirreff; Bruno Pitard; Patrick Baumhof; Francois Villinger; Philip J. Santangelo. Visualization of early events in mRNA vaccine delivery in non-human primates via PET–CT and near-infrared imaging. Nat. Biomed. Eng. 2019, 3, 371–380. https://doi.org/10.1038/s41551-019-0378-3
-
Chen, J.; Ye, Z.; Huang, C.; Qiu, M.; Song, D.; Li, Y.; Xu, Q. Lipid nanoparticle-mediated lymph node-targeting delivery of mRNA cancer vaccine elicits robust CD8+ T cell response. Proc Natl Acad Sci U S A. 2022, 23, 119(34), e2207841119. https://doi.org/10.1073/pnas.2207841119
-
Parhiz, H.; Brenner, J.S.; Patel, P.N.; Papp, T.E.; Shahnawaz, H.; Li, Q.; Shi, R.; Zamora, M.E.; Yadegari, A.; Marcos-Contreras, O.A.; Natesan, A.; Pardi, N.; Shuvaev, V.V.; Kiseleva, R.; Myerson, J.W.; Uhler, T.; Riley, R.S.; Han, X.; Mitchell, M.J.; Lam, K.; Heyes, J.; Weissman, D.; Muzykantov, V.R. Added to pre-existing inflammation, mRNA-lipid nanoparticles induce inflammation exacerbation (IE). J Control Release. 2022, 344, 50-61. https://doi.org/10.1016/j.jconrel.2021.12.027
-
Dirisala, A.; Li, J.; Gonzalez-Carter, D.; Wang, Z. Editorial: Delivery systems in biologics-based therapeutics. Front. Bioeng. Biotechnol. 2023, 11, 1274210. https://doi.org/10.3389/fbioe.2023.1274210
-
-
Expand the full forms of abbreviations or acronyms. For example, Page 6, Line 173: CRISPR/Cas9 --> clustered regularly interspersed short palindromic repeats (CRISPR)/CRISPR-associated protein 9 (Cas9)
-
Appreciated. The necessary corrections have been made.
-
Redundant abbreviations or acronyms were observed. They will increase the word count but not the scientific information. Abbreviate or acronym after their first appearance in the manuscript. For example: Lines 187, 188, and 651: Plasmid DNA (pDNA) Lines 154, and 155:Oligodeoxynucleotides (ODNs) Table 2 and Line 896: superparamagnetic iron oxide nanoparticles (SPIONs)
-
Appreciated. The necessary corrections have been made.
REVIEWER 1.
-
Abstract: In the abstract, the authors could include specific details about the current state of gene therapy, such as the number of approved products and the advancements made in the field. Additionally, it might be beneficial to mention the significance of the various categories of treated diseases.
-
Appreciated. The necessary corrections have been made.
-
Line 12-20
Abstract: In the realm of gene therapy, the pivotal moment arrived with Paul Berg's groundbreaking identification of the first recombinant DNA in 1972. This achievement set the stage for future breakthroughs. Conditions once considered insurmountable, like melanoma, pancreatic cancer, and a host of other ailments now are being addressed at their root cause — the genetic level. Presently, the gene therapy landscape stands adorned with 22 approved in-vivo and ex-vivo products including Imlygic, Luxturna, Zolgensma, Spinraza, Patisiran, and many more. In this comprehensive exploration, we delve into a rich assortment of 16 drugs, from siRNA, miRNA, and CRISPR/Cas9 to DNA aptamers and TRAIL/APO2L, as well as 46 carriers, from AAV, AdV, LNPs, and exosomes to naked mRNA, sonoporation, and magnetofection. The article also discusses the advantages and disadvantages of each product and vector type, as well as current challenges faced in the practical use of gene therapy and future potential.
-
Introduction: it could be restructured to present the historical evolution of gene therapy and the achieved milestones in a more coherent manner. The current approach seems fragmented, and integrating key events, like the approval of Kymriah, more seamlessly could enhance the narrative.
-
Appreciated. The necessary corrections have been made.
-
Line 42-68
Delving into the annals of gene therapy's history unveils a narrative shaped by pioneering discoveries and watershed moments. Frederick Griffith's 1928 experiment resonates as a seminal exploration, elucidating the transference of genetic information among bacteria through the transformative process. The monumental breakthrough arrived in 1972 with Paul Berg's revelation of the first recombinant DNA, a milestone that would reverberate across the scientific community. Berg's paradigm-shifting achievement garnered him the 1980 Nobel Prize in Chemistry, underscoring the transformative power of recombining DNA molecules. In a pivotal moment on January 19, 1989, Dr. James A. Wyngaarden, director of the National Institutes of Health (NIH), granted approval for the inaugural clinical protocol, entailing the integration of a foreign gene into immune cells for cancer patients. September 14, 1990, etches another indelible moment as W. French Anderson and his NIH colleagues conducted the first sanctioned gene therapy procedure, a life-altering intervention for a four-year-old afflicted with severe combined immunodeficiency (SCID). From 1990 to 2000, the landscape burgeoned with promise as approximately 300 clinical gene therapy trials embraced around 3000 individuals, marking a dynamic era of exploration. The march of progress ventured eastward as China's State FDA granted approval for Gendicine in 2003 to address Carcinoma, and Oncorine in 2005 for Nasopharyngeal cancer. Russia followed suit, gaining approval for Neovasculgen in 2011 to combat Atherosclerotic Peripheral Arterial Disease (PAD). The historical trajectory surged ahead, with the European FDA approving Defibrotide in 2013 for addressing SOS/VOD with multiorgan dysfunction. In 2015, the USA FDA's imprimatur ushered in a new era with IMLYGIC, ratified to treat Melanoma and Pancreatic Cancer. The progression was resounding, as 2023 bore witness to the culmination of these endeavors, witnessing the approval of 22 gene therapy products by entities such as USA FDA, EU FDA, Chinese State FDA, and the Russian Ministry of Healthcare, emblematic of the multifaceted strides taken to harness the potential of gene therapy on a global scale.
-
In line 32-33 the authors could clarify which diseases are involved and how exactly gene therapy tackles their root causes.
-
Appreciated. The additional content has been included as advised.
-
Line 29-37:
Gene therapy has become a rapidly growing field with significant advancements in recent years. This innovative therapeutic approach is revolutionizing the treatment of various diseases, including melanoma, pancreatic cancer, retinal dystrophy, spinal muscular atrophy, polyneuropathy, hereditary transthyretin-mediated amyloidosis, head and neck squamous cell carcinoma, atherosclerotic peripheral arterial disease, critical limb ischemia, nasopharyngeal cancer, SOS/VOD with multiorgan dysfunction, bacillus calmette-guérin (BCG)-unresponsive non-muscle invasive bladder cancer, hemophilia B, aromatic L-amino acid decarboxylase deficiency, multiple myeloma, cerebral adrenoleukodystrophy, lymphoma, ADA-SCID, large B-cell lymphoma, acute lymphoblastic leukemia, beta-thalassemia, leukodystrophy metachromatic by addressing the root cause of the condition — the genetic level.
-
In line 62-64 they could provide an idea of what approximate percentage of combinations actually succeed and whether there are known examples of successful cases that can support this statement.
-
Appreciated. The additional content has been included as advised.
-
Line 80:
Only about 11,5% of these combinations succeed
Line 1923:
Mullard, A. Parsing clinical success rates. Nat Rev Drug Discov. 2016, 15, 447. https://doi.org/10.1038/nrd.2016.136
-
In paragraph 2.2 I recommend to the authors this reference that I think is important and that can be of great help when modifying the manuscript (doi: https://doi.org/10.3390/jcm9123853).
-
Appreciated. The additional content has been included as advised.
-
Line 122-128:
MicroRNAs (miRNAs) offer a key advantage in gene therapy due to their small size and manipulability. Moreover, around 12 miRNAs have been identified for suppressing endogenous CFTR mRNA expression in the Caco-2 cell line. CFTR, responsible for the monogenic autosomal recessive Cystic Fibrosis (CF), impacts 1 in 3500 global live births. Maria V. Esposito et al. examined 706 CF carriers, revealing undiagnosed CFTR-RD among a subset. Genetic testing scanning analysis aids in CFTR-RD identification, offering potential for tailored follow-up and therapies to enhance outcomes. -
Line 1924-1930
Gillen, A.E.; Gosalia, N.; Leir, S.H.; Harris, A. MicroRNA regulation of expression of the cystic fibrosis transmembrane conductance regulator gene. Biochem J. 2011, 438(1), 25-32. https://10.1042/BJ20110672 -
De Palma, F.D.E.; Raia, V.; Kroemer, G.; Maiuri, M.C. The Multifaceted Roles of MicroRNAs in Cystic Fibrosis. Diagnostics (Basel). 2020, 10(12), 1102. https://10.3390/diagnostics10121102
-
Esposito, M.V.; Aveta, A.; Comegna, M.; Cernera, G.; Iacotucci, P.; Carnovale, V.; Taccetti, G.; Terlizzi, V.; Castaldo, G. Extensive CFTR Gene Analysis Revealed a Higher Occurrence of Cystic Fibrosis Transmembrane Regulator-Related Disorders (CFTR-RD) among CF Carriers. J. Clin. Med. 2020, 9, 3853. https://doi.org/10.3390/jcm9123853
-
Conclusion: The conclusion could be broadened to encompass a wider perspective on the future importance of gene therapy and areas where further improvements are possible. This would provide a stronger closing and reinforce the article's main message.
-
Appreciated. The necessary corrections have been made.
-
Line 1386-1394:
Gene therapy emerges as a robust and versatile tool poised to catalyze the advancement of personalized medicine, revolutionizing therapy efficacy and ameliorating the collateral effects of widely used pharmaceutical agents. A most rapid evolution of personalized medicine is witnessed in the realms of cancer treatment and hereditary disorders, where gene therapy strategies and products play a pivotal role. However, the potential of gene therapy extends far beyond these domains, encompassing a diverse array of pathologies, such as neurodegenerative ailments, immune disorders, inflammatory processes, and more, beckoning for further investigation.
Line 1401-1408:
Moreover, certain ethical, regulatory, and commercialization challenges must be addressed for gene therapy to become more widely adopted. Researchers embarking on the nascent phases of product development are tasked with a multifaceted consideration encompassing current technological frontiers, product commercialization, and the potential for costly solutions. While some of these innovative solutions hold transformative promise, their accessibility for patients could be constrained by elevated costs. It is imperative that these pioneering commercial decisions are underpinned by judicious forethought, as ill-conceived approaches could jeopardize not only production but also the integration of gene therapy products into medical practice. Given the potentially transformative impact of gene therapy, it is critical that these challenges are met with thoughtful and comprehensive solutions.
-
Overall, the text could benefit from greater clarity and a more consistent structure. Some passages appear complex and might be simplified to enhance comprehension.
-
Thank you for your feedback. We sincerely hope that the clarifications and revisions provided during the review process will enhance the content of the article. Given the targeted audience of specialized experts, we believe that the incorporation of more intricate details will not pose challenges to their comprehension of the more complex material. The presentation of more detailed information is anticipated to serve as a valuable asset for their professional endeavors.
-
References: Ensure that all references and citations are correctly formatted according to the journal.
-
Thank you for your diligence in reviewing the references. They have been duly checked and adjusted as needed.
-
Grammar and Syntax Review:Thoroughly check the article for any grammatical, syntactical, or punctuation errors that could compromise overall clarity.
-
Thank you. The article has been reviewed and necessary adjustments have been made to maintain its linguistic precision.
We anticipate that the refined manuscript aligns more closely with the standards of MDPI Pharmaceuticals, yet remain receptive to any additional feedback that may arise. We extend our gratitude for your attentive consideration and valued insights.
Awaiting your response to our revisions, we remain.
Best regards,
On behalf of the team of authors,
Aladdin Y. Shchaslyvyi

Round 2
Reviewer 1 Report
I believe that the study has sufficient merit to be considered for publication
Author Response
Thank you for your assessment.
We greatly appreciate your support and consideration for the publication of our study.
Best regards,
On behalf of the team of authors,
Aladdin Y. Shchaslyvyi
Institute of Molecular Biology and Genetics,
National Academy of Sciences of Ukraine, Kyiv, Ukraine

Reviewer 2 Report
The authors addressed the comments and suggestions raised by the reviewer.
Minor issue: Onpattro is a lipid nanoparticle (LNP), not a liposome. Please confirm the above.
Author Response
Thank you for your comment.
We have confirmed and edited the manuscript accordingly.
Best regards,
On behalf of the team of authors,
Aladdin Y. Shchaslyvyi
Institute of Molecular Biology and Genetics,
National Academy of Sciences of Ukraine, Kyiv, Ukraine
